# Chp1 is a dedicated chaperone at the ribosome that safeguards eEF1A biogenesis

Melania Minoia[1,7], Jany Quintana-Cordero [1,7], Katharina Jetzinger[1,2], Ilgin Eser Kotan [2], Kathryn Jane Turnbull [3,4], Michela Ciccarelli[1], Anna E. Masser[1], Dorina Liebers[2], Eloïse Gouarin[1], Marius Czech[1], Vasili Hauryliuk [5,6], Bernd Bukau [2], Günter Kramer[2] & Claes Andréasson [1] ✉

Cotranslational protein folding depends on general chaperones that engage highly diverse nascent chains at the ribosomes. Here we discover a dedicated ribosome-associated chaperone, Chp1, that rewires the cotranslational folding machinery to assist in the challenging biogenesis of abundantly expressed eukaryotic translation elongation factor 1A (eEF1A). Our results indicate that during eEF1A synthesis, Chp1 is recruited to the ribosome with the help of the nascent polypeptide-associated complex (NAC), where it safeguards eEF1A biogenesis. Aberrant eEF1A production in the absence of Chp1 triggers instant proteolysis, widespread protein aggregation, activation of Hsf1 stress transcription and compromises cellular fitness. The expression of pathogenic eEF1A2 variants linked to epileptic-dyskinetic encephalopathy is protected by Chp1. Thus, eEF1A is a difficult-to-fold protein that necessitates a biogenesis pathway starting with dedicated folding factor Chp1 at the ribosome to protect the eukaryotic cell from proteostasis collapse.

Products of aberrant protein synthesis constitute a significant burden to the cellular proteostasis network[1]. Polypeptides that emerge from the ribosomal exit tunnel have not yet attained their native structure and are susceptible to misfolding, aggregation and premature degradation[2]. Proteolytic systems target nascent chains with properties that impede cotranslational folding, including high translation rate, large domains, elevated hydrophobicity and high aggregation propensity[3]. The folding is further challenged by destabilizing mutations, gene expression errors and stress-induced protein misfolding[2,4]. Productive protein biosynthesis therefore relies on mechanisms that ensure the efficient folding of nascent proteins to safeguard proteostasis.

Ribosome-associated chaperones provide a key mechanism that promotes folding and protects nascent polypeptides from premature degradation. These chaperones associate with the ribosome near the polypeptide exit tunnel to bind a broad range of nascent polypeptides. Among these chaperones are a customized Hsp70/J-domain protein chaperone system termed the ribosome-associated complex (RAC), the ribosome-associated Hsp70 chaperone, Ssb and the nascent polypeptide-associated complex (NAC)[4–6]. NAC is a heterodimer composed of the α- (Egd2) and β- (Egd1 or Btt1) subunits[7–9]. Genetic removal of NAC leads to increased ubiquitylation of newly synthesized polypeptides[3,7,10]. NAC transiently captures the signal recognition particle (SRP) which allows it to scan nascent chains while

[1]Department of Molecular Biosciences, The Wenner-Gren Institute, Stockholm University, Stockholm, Sweden. [2]Center for Molecular Biology of the University of Heidelberg (ZMBH), DKFZ-ZMBH Alliance, Heidelberg, Germany. [3]Department of Clinical Microbiology, Rigshospitalet, 2200 Copenhagen, Denmark. [4]Department of Molecular Biology, Laboratory for Molecular Infection Medicine Sweden, Umeå Centre for Microbial Research, Science for Life Laboratory, Umeå University, Umeå, Sweden. [5]Science for Life Laboratory, Department of Experimental Medical Science, Lund University, Lund, Sweden. [6]University of Tartu, Institute of Technology, 50411 Tartu, Estonia. [7]These authors contributed equally: Melania Minoia, Jany Quintana-Cordero. ✉e-mail: claes.andreasson@su.se

simultaneously preventing SRP from binding to ribosomes that do not expose signal sequences[11]. As a consequence, NAC is a negative regulator of protein translocation to the endoplasmic reticulum and a potent suppressor of protein aggregation[5,12]. The role that NAC plays to coordinate the recruitment of nascent-chain interacting factors other than SRP is less clear.

Cellular protein production requires a substantial investment in the highly abundant protein component of the translation machinery itself. The eukaryotic translation elongation factor 1A (eEF1A) is among the most abundant proteins in the cell, accounting for 1.8% of the total protein content in yeast[13]. eEF1A delivers aminoacylated-tRNA (aa-tRNA) to the A site of the ribosome for decoding of mRNA. It has three domains, including an N-terminal GTPase domain (domain I) that allosterically controls its interactions with aa-tRNA, the ribosome and the guanine nucleotide exchange factor[14,15]. Analysis of protein synthesis rates and ribosome occupancies show that eEF1A belongs to the group of proteins with the highest synthesis rates[16]. The mechanism that ensures the folding of this highly expressed multidomain GTPase has just started to uncover. Recently, Zpr1-Aim29 was identified as a specialized chaperone-cochaperone system that assists the biogenesis of eEF1A[17,18]. Zpr1 uses its zinc-finger and alpha-helical hairpin structures to facilitate folding of newly synthesized eEF1A through a mechanism that requires client GTP hydrolysis. The Aim29 cochaperone binds the Zpr1-eEF1A complex in its GTP-bound form and drives

eEF1A release from Zpr1. The identification of Zpr1 and Aim29 as biogenesis factors for eEF1A opens up the possibility that eEF1A production depends on a dedicated pathway involving multiple chaperones.

Here, we show that the previously uncharacterized protein Chp1 is a ribosome-associated chaperone dedicated to the folding of nascent eEF1A. Chp1 transiently binds the GTPase domain of eEF1A during its synthesis. NAC forms a complex with Chp1 and stabilizes the interactions with ribosome-eEF1A nascent chain complexes. In addition to its role in biosynthesis, Chp1 binds and stabilizes fully-synthesized eEF1A that harbor destabilizing mutations in the GTPase domain from patients with epileptic-dyskinetic encephalopathy. Overall, Chp1 secures the faithful biosynthesis of eEF1A thereby protecting the proteostasis system from the damage caused by aberrant production of this highly expressed translation factor.

## Results

### Chp1 interacts with translating ribosomes and forms a complex with NAC

To identify novel ribosome-associated chaperones that safeguard proteostasis, we compared a dataset of ribosome-associated proteins (RAP) in yeast with previously defined gene deletions that trigger the activation of the proteostasis stress sensor Hsf1 (Fig. 1a)[19,20]. The top-Hsf1 activating genes encoded well-characterized chaperones, including components of the Hsp70 and Hsp90 systems. Interestingly the

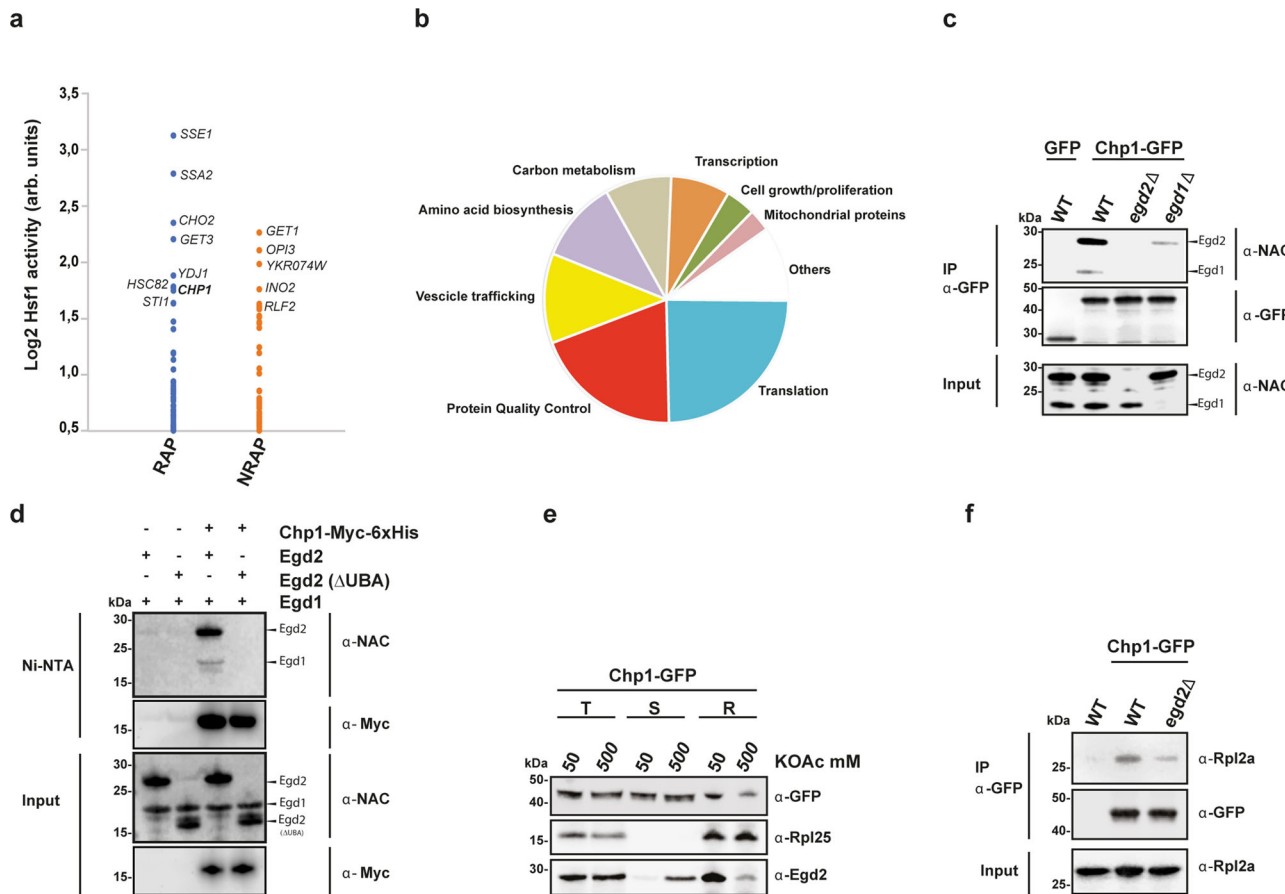

**Fig. 1 | Chp1 binds ribosomes aided by direct interaction with the α-subunit of NAC. a** Hsf1 activity in yeast strains with single deletions of genes encoding either ribosome-associated proteins (RAP, blue) or non-ribosome associated proteins (NRAP, orange). **b** Analysis of functional categories of Chp1-interacting proteins identified by in vivo site-specific UV crosslinking followed by two-step affinity purification and peptide mass fingerprinting identification using LC-MS. **c** Chp1-GFP and NAC subunits co-IP from the indicated yeast strains. The experiment was performed three times. **d** Western blot analysis of Ni-NTA matrix binding of Chp1-

Myc-6His mixed with NAC (Egd1- Egd2) or NAC lacking the UBA domain (Egd1- Egd2 ΔUBA). The experiment was performed three times. **e** Total soluble yeast cell lysate (T) was separated into ribosomal (R) and soluble (S) fractions in the presence of either 50 mM or 500 mM KOAc and the localization of Chp1-GFP, Egd2 and Rpl25 among the different fractions was analyzed by western blot. The experiment was performed three times. **f** Chp1-GFP and Rpl2A co-IP from the indicated yeast strains. The experiment was performed three times.

uncharacterized open reading frame (ORF) YPL225w, here named *CHP1* (Chaperone 1 for eEF1A), was associated with one of the strongest activations of Hsf1 among all the RAP. As evidenced by structural comparison, *CHP1* encodes a highly conserved eukaryotic protein, yet, the function of Chp1 and its orthologs, including the human ortholog PBDC1, remains to be elucidated (Supplementary Fig. 1a). To identify interaction partners of Chp1 we used in vivo site-specific UV cross-linking. Position Leu12 of Chp1 was replaced with the UV-activatable amino acid p-benzoyl-l-phenylalanine (Bpa) via stop codon suppression, followed by two-step affinity purification and peptide mass fingerprinting identification using LC-MS. The Bpa substitution at position Leu12 was chosen after testing a number of conserved residues (Supplementary Fig. 1b). 24% and 19% of the identified Chp1-crosslinking partners were found to be involved in translation and protein quality control, respectively (Fig. 1b, Supplementary Data 1). The α-subunit of NAC, Egd2, was ranked as a top hit and using serum detecting Egd1 and Egd2 we found that NAC directly crosslinked with Chp1-L12Bpa (Supplementary Fig. 1c). We performed structural modeling of a complex between Chp1 and NAC and the human homologues using ColabFold. Despite that the models did not recapitulate all 12 strands of the β-barrel-like heterodimer determined by crystallography for the human NAC heterodimerization domain[21,22] they both suggested that Chp1 binds NAC via direct interaction with the UBA domain of its α-subunit, Egd2 (Supplementary Fig. 1d and Supplementary Data 2)[23]. Co-immunoprecipitation (co-IP) of GFP-tagged Chp1 from yeast cells and purification of Chp1 from *E. coli* co-expressing NAC (Egd1 and Egd2) validated that Chp1 forms a complex with NAC via direct binding to Egd2 (Fig. 1c and Supplementary Fig. 1e). Using purified components, we confirmed a direct interaction between Chp1 and NAC that was dependent on the presence of the UBA domain (Fig. 1d). Hence, Chp1 directly interacts with NAC via Egd2 dependent on the UBA domain.

NAC resides at the nascent polypeptide tunnel exit[4,5]. We confirmed by sedimentation experiments and co-IP that Chp1 associates with the ribosome (Fig. 1e, f and Supplementary Fig. 1f). In line with previous studies on NAC[24,25], high-salt washes reduced the interaction not only of NAC but also of Chp1 (Fig. 1e). The association with the ribosome was reduced when *EGD2* was deleted (Fig. 1f and Supplementary Fig. 1f). Polysome profile analysis revealed that nascent chain release upon puromycin treatment abrogated the interaction of Chp1 with translating ribosomes, demonstrating the key role of the nascent polypeptides in Chp1 recruitment to the 80 S ribosomes (Supplementary Fig. 1g). The data taken together indicate that Chp1 interacts with translating ribosomes in a nascent chain dependent manner and that this interaction is facilitated by NAC.

## Chp1 binds the nascent GTPase domain of eEF1A with the help of NAC

We performed selective ribosome profiling (SeRP) to find Chp1 interactors at ribosome-nascent chain complexes (RNCs) on a proteome-wide scale[26,27] (Fig. 2a). The most enriched transcripts identified by Chp1-SeRP belonged to the near-identical *TEF1* and *TEF2* (*TEF1/2*), encoding eEF1A, suggesting that eEF1A is the sole nascent-chain substrate of Chp1 (Fig. 2b and Supplementary Fig. 2a). The interaction between Chp1 and ribosome-eEF1A nascent chain complexes was highly specific since transcripts encoding homologous translation-associated GTPases such as *EFT1/2*, *SUP35*, *TUF1* and *HBS1* were not enriched in the Chp1-bound translatome. Analysis of the Chp1 enrichment profile along the *TEF1/2* transcripts showed that Chp1 engaged the eEF1A nascent chain after translation of the first 100 codons, fortified its association during the translation of the next 50 codons and finally disengaged after translation of approximately 270 codons (Fig. 2c). Taking into consideration the polypeptide exit tunnel length, which can accommodate 30 amino acids, the data indicate that Chp1 first engages nascent eEF1A when the N-terminal 70 amino acids

of the GTPase domain (domain I) have emerged from the ribosomal tunnel and stays associated until the entire domain has been exposed.

The SeRP data raised the possibility that Chp1 forms a complex with the GTPase domain of eEF1A during its synthesis. Consistently, structural modeling of a Chp1-GTPase domain complex using Colab-Fold indicated that Chp1 binds a motif within the N-terminal 70 amino acids of the GTPase domain (Fig. 2d and Supplementary Data 3). The model predicts that Chp1 uses its protruding N-terminal α-helix to interact with the GTPase domain α-helices α1, α2 and α3, which cover residues 20 to 69. We directly tested whether Chp1 and the isolated domain I of eEF1A associate. Indeed, purified Chp1 and eEF1A domain I formed a complex in vitro (Fig. 2e). This interaction was also apparent as eEF1A domain I specifically copurified with Chp1 upon co-expression in the heterologous host *E. coli* (Fig. 2f). In addition, size-exclusion chromatography analysis of the copurified fraction from the *E. coli* recombinant system confirmed the presence of a stable Chp1-eEF1A domain I complex (Supplementary Fig. 2b). Interestingly, coexpression of Chp1 with the GTPase domain in *E. coli* cells increased the expression levels of the eEF1A domain (Fig. 2g–i). Having established the Chp1-eEF1A domain I complex, we used in vitro site-specific UV crosslinking between purified Chp1 with single Bpa substitutions in the N-terminal α-helix (Q18, V21 or E25) and domain I of eEF1A to test the involvement of this helix in the interaction (Supplementary Fig. 2c). All three positions harboring the zero-spacer crosslinker Bpa formed specific crosslinks with eEF1A domain I, demonstrating direct interaction with the domain (Supplementary Fig. 2d). Upon mutational analysis we found using the heterologous coexpression system that alanine substitutions of Chp1 residues predicted to interact with eEF1A, or complete removal of residues 2 to 28 significantly decreased the interaction (Supplementary Fig. 2e). The most prominent impairment was observed for the deletion mutant, resulting in 58% reduction of the binding. Finally, using the heterologous coexpression system, we found that the first 70 amino acids of the GTPase domain (β1-α1-α2-α3) are sufficient to form a complex with Chp1 (Fig. 2j). Thus, the cotranslational interactions between Chp1 and the eEF1A GTPase domain that we detected by the SeRP are recapitulated in the ColabFold model as well as by reconstitution. According to the model, this interaction is not compatible with a nucleotide-bound conformation of the eEF1A GTPase domain. Moreover, Chp1 forms a stable complex with the GTPase domain and increases its expression level in a heterologous host. Together, this data suggest that the stability of the domain increases when bound by Chp1. Lastly, we asked if the Chp1 interaction partner, NAC, facilitates the binding of Chp1 to the nascent eEF1A. Chp1-SeRP in *nac*Δ cells (*egd1*Δ *egd2*Δ *btt1*Δ) revealed that in the absence of NAC, Chp1 still engages the nascent GTPase domain of eEF1A once it emerges from the ribosomal exit tunnel. However, Chp1 prematurely disengaged the nascent chain before the entire domain I was synthesized (Fig. 2k). This finding suggests that although Chp1 does not strictly depends on NAC for initial engagement of nascent eEF1A, NAC aids further interactions until the entire domain has been translated.

Together, our data indicate that Chp1 engages nascent eEF1A during its biosynthesis as soon as the first N-terminal 70 amino acids emerge from the polypeptide exit tunnel. Chp1 cotranslationally stabilizes the growing GTPase domain chain and NAC stabilizes the interaction. Finally, Chp1 disengages the nascent eEF1A once the complete GTPase domain is exposed out of the ribosomal tunnel, which likely leads to cotranslational folding of the domain into a stable structure.

## Chp1 binds non-natively folded eEF1A

eEF1A extensively misfolds and aggregates in the *E. coli* expression system[17], likely a consequence of the absence of suitable biogenesis factors. Consistent with previous reports we found that full-length eEF1A quantitatively aggregated when expressed in *E. coli*

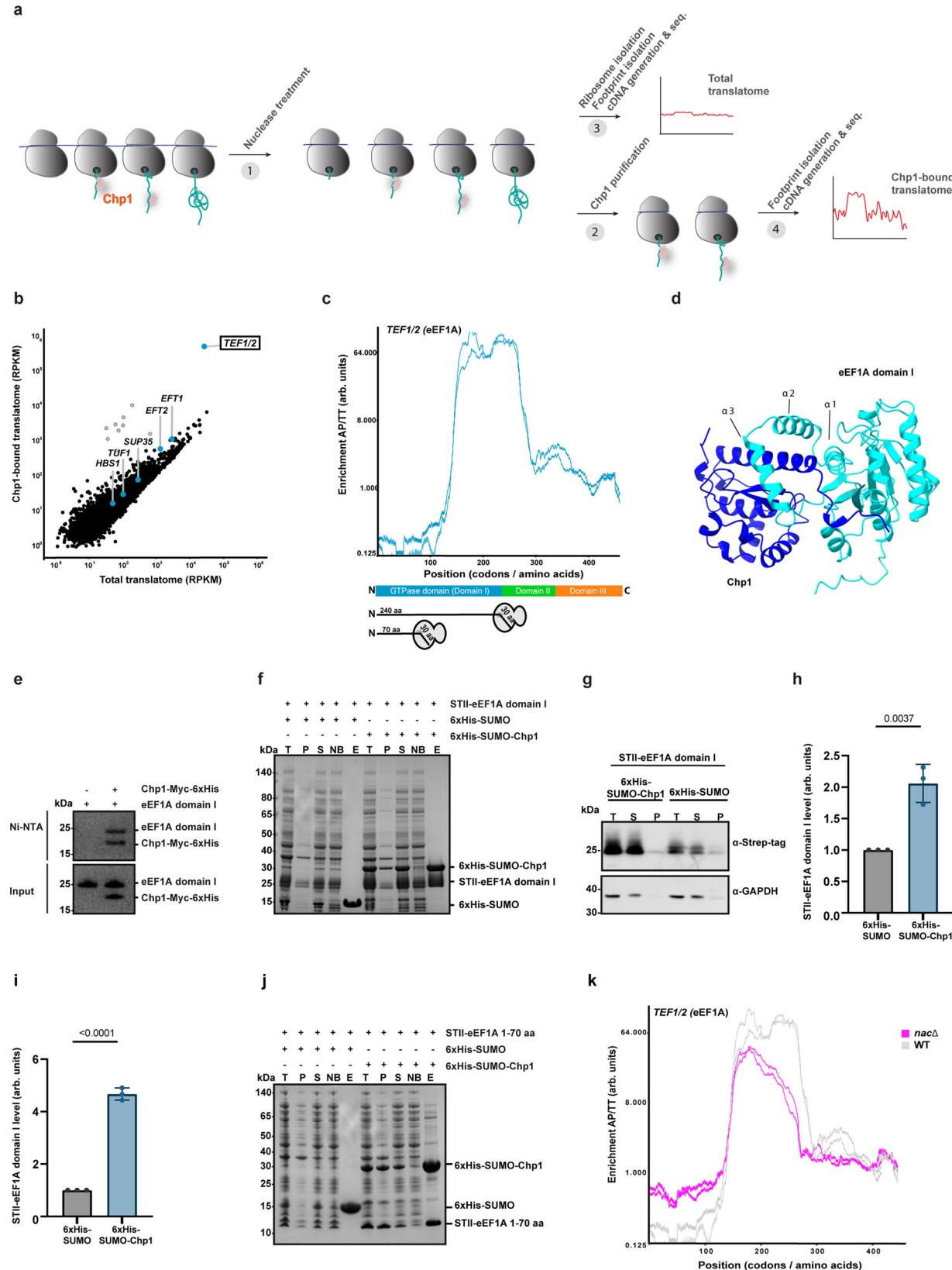

(Supplementary Fig. 3a). In contrast, when Chp1 was coexpressed, eEF1A solubility was significantly increased and this soluble fraction could be copurified with Chp1 (Supplementary Fig. 3a–e). The copurified complex contained many contaminating proteins not observed in the control purifications suggesting that eEF1A was misfolded and unspecifically interacted with these species (Supplementary Fig. 3a).

Chp1 itself was expressed as a highly soluble protein in the heterologous system. However, when coexpressed with eEF1A, Chp1 was dragged into the pellet fraction of the lysate demonstrating an interaction with misfolded eEF1A, presumably the result of exposed Chp1 interaction sites on the non-natively folded GTPase domain. While Chp1 interacts with misfolded eEF1A, we predicted it should not bind

**Fig. 2 | Chp1 interacts with the nascent GTPase domain (domain I) of eEF1A at the ribosome with the help of NAC. a** Workflow of Chp1-SeRP[26]. **b** Transcript abundance in reads per kilobase million (RPKM) from Chp1-SeRP ($n = 2$). Non-specifically bound transcripts in light grey. **c** Chp1 interaction profile along eEF1A nascent chain determined by Chp1-SeRP ($n = 2$). Nascent chains of 100/270 residues with 30 residues in the ribosomal tunnel are depicted. The coverage of eEF1A domains along the protein length is represented (domain I - blue, domain II - green, domain III - orange). **d** Ribbon diagram of ColabFold model showing the predicted complex between Chp1 and the domain I of eEF1A (Chp1 - dark blue, domain I of eEF1A – cyan). Helixes α1, α2 and α3 (20–69 residues in the GTPase domain of eEF1A) are labeled. **e** SDS-PAGE (Coomassie Brilliant Blue) analysis of Ni-NTA matrix binding of Chp1-Myc-6His mixed with eEF1A domain I. The experiment was performed three times. **f** STII-eEF1A domain I was coexpressed with 6xHis-SUMO-Chp1 or 6xHis-SUMO in *E. coli*. Total cell lysate (T) was separated into pellet (P) and soluble (S) fractions and the S fraction was subjected to IMAC purification. NB (non-bound) and E (eluted) fractions from the IMAC purification. All fractions were analyzed by SDS-PAGE followed by coomassie staining. The experiment was performed three times. **g** Total cell lysate (T) from *E. coli* cells coexpressing STII-eEF1A domain I and 6xHis-SUMO-Chp1 or 6xHis-SUMO was separated into soluble (S) and pellet (P) and the level of STII-eEF1A domain I in the different fractions was analyzed by western blot. **h** Quantification of the level of recombinant STII-eEF1A domain I in the T fraction from (**g**) (means ± SD, $n = 3$ independent cultures, two-tailed *t* test). **i** Quantification of the level of recombinant STII-eEF1A domain I in the S fraction from (**g**) (means ± SD, $n = 3$ independent cultures, two-tailed *t*-test). **j** As in (**f**) but STII-eEF1A 1-70 aa was coexpressed with 6xHis-SUMO-Chp1 or 6xHis-SUMO. The experiment was performed three times. **k** Chp1 interaction profiles along Chp1 nascent chain determined via Chp1-SeRP of WT (grey) or *nacΔ* (magenta) yeast strains ($n = 2$).

the natively folded protein. To investigate this interaction, we performed an in vitro co-IP of Chp1-mCherry using eEF1A purified from its native source. Indeed, only minimal interaction between Chp1 and eEF1A could be observed with the highest interaction being detected in the presence of EDTA (10% of purified Chp1 was bound to eEF1A), a treatment that impedes nucleotide binding and thus destabilizes the GTPase domain[15] (Supplementary Fig. 3f). Preincubation of eEF1A with nucleotide or high glycerol concentrations, both known to stabilize the protein[15,28,29], released Chp1. In a parallel approach, in vitro site-specific UV crosslinking between Chp1 with Bpa substitution at position E25 and eEF1A demonstrated impaired interaction between the proteins in the presence of nucleotide or high glycerol concentration (Supplementary Fig. 3g). Taken together we find that Chp1 does not interact with native eEF1A but can bind its unfolded or destabilized form improving its solubility.

## Chp1 secures faithful biogenesis of eEF1A

Recently, Zpr1 has been uncovered as a chaperone that specifically assists eEF1A folding during its biosynthesis[17]. In line with a role for Chp1 in the biogenesis of eEF1A, comparative synthetic genetic array (SGA) data analysis suggested numerous shared genes that exhibit negative genetic interactions with both *CHP1* and *ZPR1*[30]. The analysis mapped eEF1A-related functions including tRNA biogenesis, aminoacylation and the structural *TEF1* itself (Supplementary Fig. 4a). Looking exclusively at the negative genetic interactions of *CHP1*, we substantiated the finding of an involvement in translation (Supplementary Fig. 4b). This was further supported by finding that 7 of the *CHP1* negative interactors involved in translation and protein quality control (*YDJ1*, J-domain protein for Hsp70; *TEF1*, eEF1A; *TEF4*, eEF1Bγ; *RSP5*, ubiquitin E3 ligase; *HYP2*, eIF5A; *RPSOA*, ribosomal 40 S subunit protein S0A; *RPT1*, ATPase 19 S proteosome) were also identified as physical interactors in our MS analysis (Supplementary Fig. 4c). Direct testing by construction of a *tef1Δ chp1Δ* mutant, revealed a more severe growth impairment at 30 °C and 37 °C compared to *tef1Δ* cells (Fig. 3a, b and Supplementary Fig. 5a, b). Also, *chp1Δ* cells displayed a mild growth impairment most clearly visible at 37 °C in the form of diauxic shift at lower cell densities but with no impact on doubling times. Introducing an extra copy of the *TEF1* gene (+*TEF1*) did not visibly impact the growth of the WT cells but suppressed the diauxic shift phenotype of *chp1Δ* cells. Thus, decreasing the gene dosage of eEF1A makes the maximal growth potential of cells sensitive to the loss of Chp1.

We quantified eEF1A levels in the mutants using western blot analysis (Fig. 3c and Supplementary Fig. 5c). In line with the growth phenotypes, we observed 36% reduction of eEF1A levels in *chp1Δ* cells, while *tef1Δ* cells exhibited 74% reduction. Hence, the lower levels of eEF1A in *chp1Δ* cells result in only mild growth phenotypes, while further reduction of the levels in *tef1Δ* cells apparently makes translation elongation rates limiting for growth. The *chp1Δ tef1Δ* cells exhibited 78% decrease of eEF1A levels and further reduced growth compared to the *tef1Δ* strain. Interestingly, while +*TEF1* cells expressed 184% eEF1A, *chp1Δ* + *TEF1* cells failed to significantly increase the *eEF1A* levels compared to *chp1Δ* cells, suggesting that eEF1A expression requires Chp1. Thus, overexpression of eEF1A increased the dependency on Chp1 for its successful production. Inactivation of NAC (*nacΔ*) alone or in combination with Chp1 caused no further reduction of the total eEF1A levels (Supplementary Fig. 5d). Since total *TEF1* and *TEF2* mRNA levels were not reduced in the *chp1Δ* cells (Supplementary Fig. 5e), we investigated whether Chp1 is required for the efficient biogenesis of eEF1A. Upon translation arrest using cycloheximide, the pre-existent eEF1A population was stable for up to 3 hours in WT as well as in *chp1Δ* cells (Fig. 3d). In contrast, when eEF1A was expressed from the induced *GAL1* promoter, the rate of eEF1A production was drastically reduced in *chp1Δ* cells, while mRNA levels were comparable in WT and *chp1Δ* cells, indicating that Chp1 plays a major role in specifically de novo protein synthesis of eEF1A (Fig. 3e). To investigate the biogenesis defect, we isolated protein aggregates and found that while the amount of ubiquitylated protein aggregates were increased in *chp1Δ* cells, eEF1A did not accumulate in the aggregates (Supplementary Fig. 5f). Additionally, ribosome profiling (RP) analysis showed the same ribosome density along the *TEF1/2* transcripts for WT and *chp1Δ* cells ruling out a defect in the translation of these transcripts in the absence of Chp1 (Supplementary Fig. 5g). These findings raised the possibility that newly synthesized eEF1A in *chp1Δ* cells is rapidly degraded by the ubiquitin-proteasome system (UPS). Indeed, inactivation of Chp1 triggered extensive polyubiquitylation of eEF1A and proteasome inhibition by Bortezomib reinforced the polyubiquitylation phenotype (Fig. 3f). Thus, Chp1 protects nascent eEF1A at the ribosome from premature proteasomal degradation.

## Aberrant biogenesis of eEF1A in *chp1Δ* cells burdens the proteostasis system

Aberrant biogenesis of eEF1A by Zpr1 depletion has been linked to translation defects and to proteostasis imbalance[17]. We used RP to investigate if the reduced eEF1A levels in *chp1Δ* cells cause global translation defects. Metagene profiles of *chp1Δ* cells showed a typical enrichment of ribosomes at the start of the coding sequences of all translated genes, followed by an overall uniform ribosome distribution, indicating that the reduced levels of eEF1A did not cause major global changes in the translation process (Supplementary Fig. 6a, b, $R = 0.99$, $p < 2.2e^{-16}$). In contrast to Zpr1 depletion, the integrated stressed response was only weakly induced as evidenced by a mere 1.5-fold increased translation of *GCN4* (Supplementary Fig. 6c). Instead, differential expression analysis of the RP data revealed that the most significantly up-regulated genes in *chp1Δ* cells were associated with proteostasis stress transcription driven by Hsf1 activation (Fig. 4a, b). Hence, the absence of Chp1 specifically leads to a proteostasis imbalance and activation of Hsf1 that upregulate among others heat shock proteins.

We found that the *chp1Δ* proteostasis phenotype was directly caused by off-pathway misfolding of eEF1A rather than indirectly via translation defects relating to limiting eEF1A levels. Deletion of *TEF1* in *chp1Δ* cells suppressed the proteostasis phenotype while increasing the expression by integrating one extra copy of *TEF1* in the genome (+*TEF1*) reinforced it. This genetic interaction profile was apparent when assessing an Hsf1-dependent transcriptional reporter as well as by the expression levels of heat shock proteins Fes1 and Btn2 (Fig. 4c–e). Extending the analysis to microscopically visualise the proteostasis phenotype at the level of protein aggregates (Hsp104-GFP) resulted in a similar genetic interaction profile (Fig. 4f, g)[31]. In line, +*TEF1* induced a pronounced growth defect in *chp1Δ* cells at the stressful temperature of 39 °C (Supplementary Fig. 7a). Assessing the genetic interactions between the NAC genes and *CHP1* supported a subordinated role for NAC in aiding Chp1 during eEF1A folding. While *chp1Δ* cells displayed a growth defect at 39 °C, *nacΔ* cells grew indistinguishable from WT cells, and the *nacΔ chp1Δ* double mutant phenocopied *chp1Δ* cells (Supplementary Fig. 7b). In the same line, unlike *chp1Δ* cells, *nacΔ* cells did not activate Hsf1 or promote cytosolic protein aggregation (Supplementary Fig. 7c–e). Overall, the data confirm that *chp1Δ* cells display a proteostasis stress phenotype and show that this phenotype directly correlates with the expression levels of eEF1A. Hence nascent eEF1A that fails to fold in the absence of Chp1 significantly burdens the proteostasis system and causes the proteostasis phenotype.

## Chp1 binds and stabilizes eEF1A F98C missense variant

Since our data indicate that in addition to its ribosome-associated role during the biogenesis of eEF1A, Chp1 has the potential to bind

misfolded full-length translation factor, we set out to investigate whether Chp1 is involved in the quality control of destabilized pathogenic eEF1A variants. Sequence variants in *EEF1A2* have been found in patients with neurodevelopmental epilepsy and intellectual disability[32–34] (Fig. 5a and Supplementary Fig. 8a). We expressed WT eEF1A (V5-*TEF2*) as well as mutants representing nine of the polymorphisms associated to epileptic-dyskinetic encephalopathy (G70S, D91N, F98C, M102V, A125E, R264W, P331L, G382R and R421C) in WT and *chp1Δ* yeast cells. All mutants but one (R421C) exhibited significantly reduced protein levels in WT cells with even further reduced levels in *chp1Δ* cells (Supplementary Fig. 8b). The mutants appeared to represent recessive destabilized variants, since they did not cause prominent growth impairment (Supplementary Fig.8c, d). We tested the ability of Chp1 to interact with each variant by co-IP. As expected, we observed only a weak interaction between WT eEF1A and Chp1. Mutants G70S, D91N, A125E, R264W and G382R displayed even further reduced interaction with Chp1. Interestingly, mutants F98C and M102V, which are changes buried within the switch II helix of the GTPase domain important for nucleotide interactions, showed a 3-fold increase in Chp1 binding (Fig. 5b). We assessed the biogenesis rate and stability of two variants (G70S and F98C) selected based on the location of mutations (Switch I and II of the GTPase domain, respectively) and the difference in the interaction with Chp1. When G70S and F98C eEF1A variants were expressed from the induced *GAL1* promoter, the rate of their production was significantly reduced in *chp1Δ* cells, indicating that Chp1 plays a major role in de novo synthesis of also the eEF1A variants (Fig. 5c, d and Supplementary Fig. 8e). Upon translation arrest, the F98C variant was found to be significantly less stable than

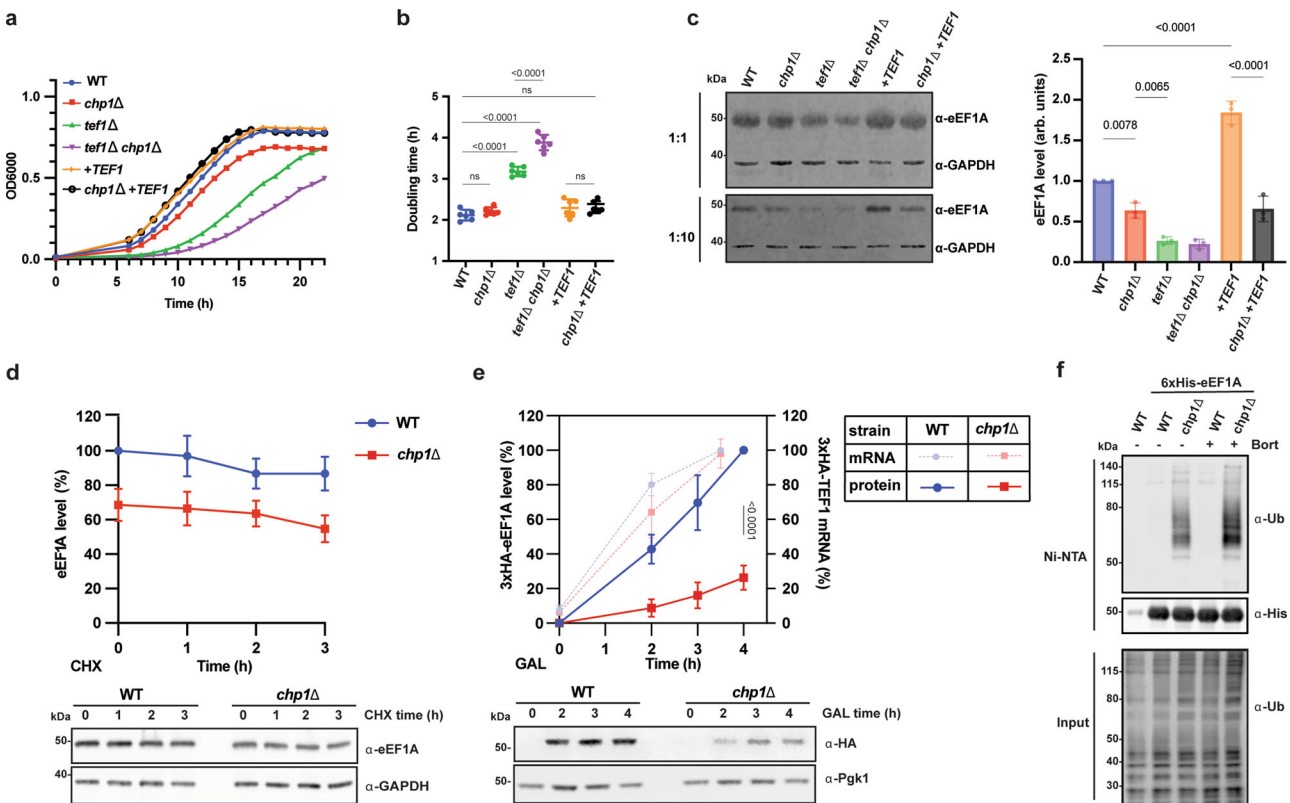

**Fig. 3 | Chp1 safeguards the biogenesis of eEF1A. a** Growth curves of the indicated yeast strains at 37 °C (means, *n* = 4). **b** Doubling times of the yeast strains in **a** (means ± SD, *n* = 6, one-way ANOVA). **c** Expression levels of eEF1A in the indicated yeast strains. Western blot analysis of 0.12 and 0.012 OD600 units of cell culture are shown. Quantifications of the samples represent means ± SD (*n* = 3), one-way ANOVA. **d** eEF1A level at the indicated times following translation arrest by cycloheximide (CHX) treatment (means ± SD, *n* = 5 independent cultures). **e** De novo

expression of 3xHA-eEF1A (3xHA-*TEF1*) from the *GAL1* promoter after galactose (GAL) induction. Solid lines correspond to protein level measurements by western blot analysis (means ± SD, n = 5, two-tailed *t* test). Dashed-lines correspond to qPCR measurements of mRNA levels (means ± SD, *n* = 4). **f** Polyubiquitylation of eEF1A (6xHis-Tef2) purified under denaturing conditions from yeast cells treated with 1 mM Bortezomib ( + Bort) 15 minutes before harvest. Experiment was performed three times.

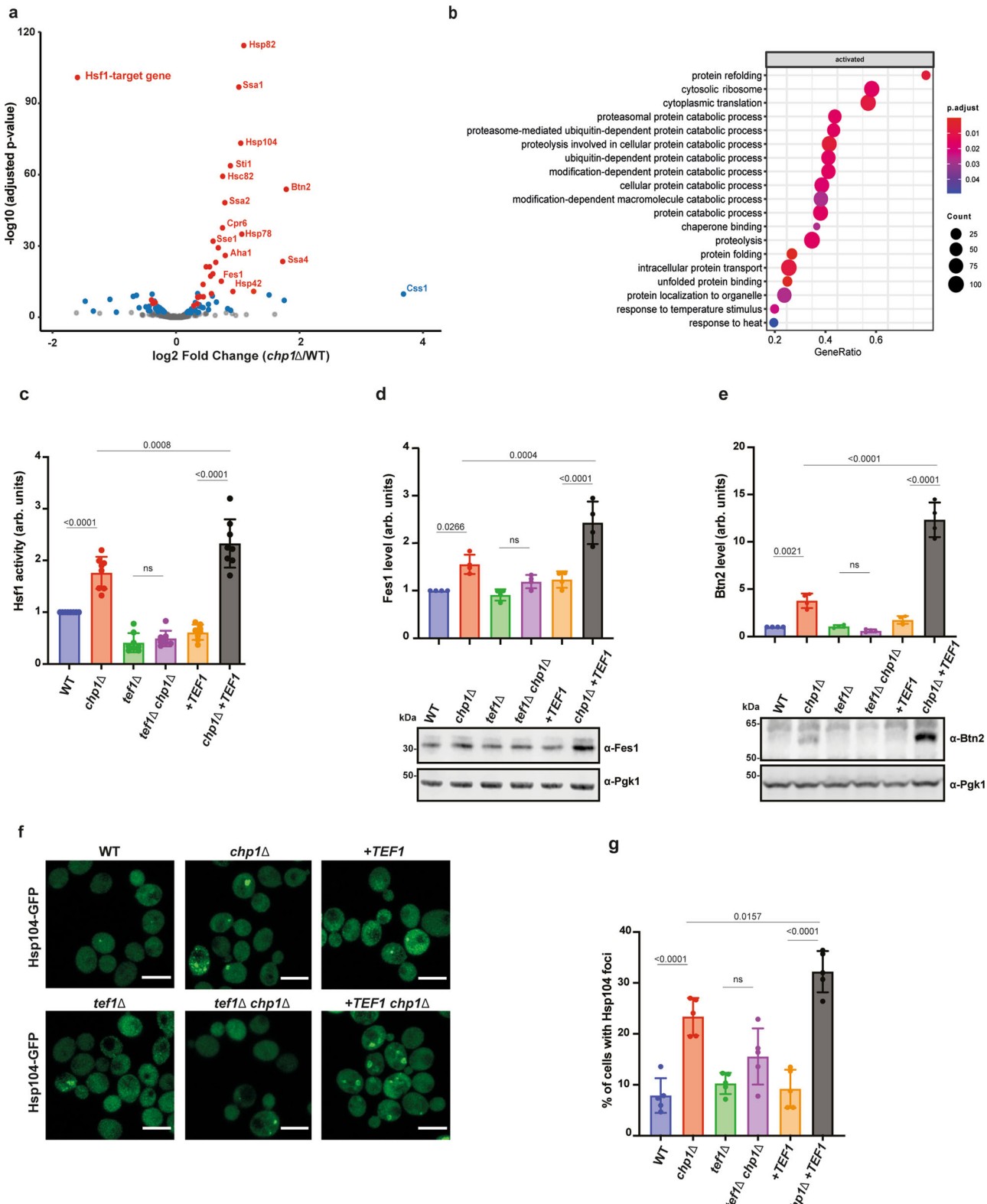

**Fig. 4 | Aberrant eEF1A biogenesis in *chp1Δ* cells severely burdens the proteostasis system. a** Differential gene expression obtained by comparing the translatome of *chp1Δ* and WT cells. Hsf1-target genes with adjusted *p*-value < 0.01 (Wald test corrected for multiple hypotheses by the Benjamini and Hochberg method) are shown in red and non-Hsf1 targets in blue. **b** Metascape Enrichment Analysis indicating significantly enriched GO terms within the detected differentially expressed genes in **a** using one-sided Fisher's exact test corrected as in **a**. **c** Hsf1 activity in WT, *chp1Δ*, *tef1Δ*, *tef1Δ chp1Δ*, *+TEF1* and *chp1Δ+TEF1* yeast strains determined by the $P_{CYC1\text{-}HSE}$-yNluc bioluminescent reporter (means ± SD, *n* = 8, one-way ANOVA). **d** Expression level of Fes1 in WT, *chp1Δ*, *tef1Δ*, *tef1Δ chp1Δ*, *+ TEF1* and *chp1Δ +TEF1* yeast strains determined by western blot analysis (means ± SD, *n* = 4, one-way ANOVA). **e** As in (**d**) but expression level of Btn2 was analyzed (means ± SD, *n* = 4, one-way ANOVA). **f** Micrographs of Hsp104-GFP expressed in WT, *chp1Δ*, *tef1Δ*, *tef1Δ chp1Δ*, *+ TEF1* and *chp1Δ +TEF1* yeast strains. Scale bar is 3 μM. **g** Quantification of the fraction of cells with Hsp104-GFP foci in (**f**) (means ± SD, *n* = 5, one-way ANOVA).

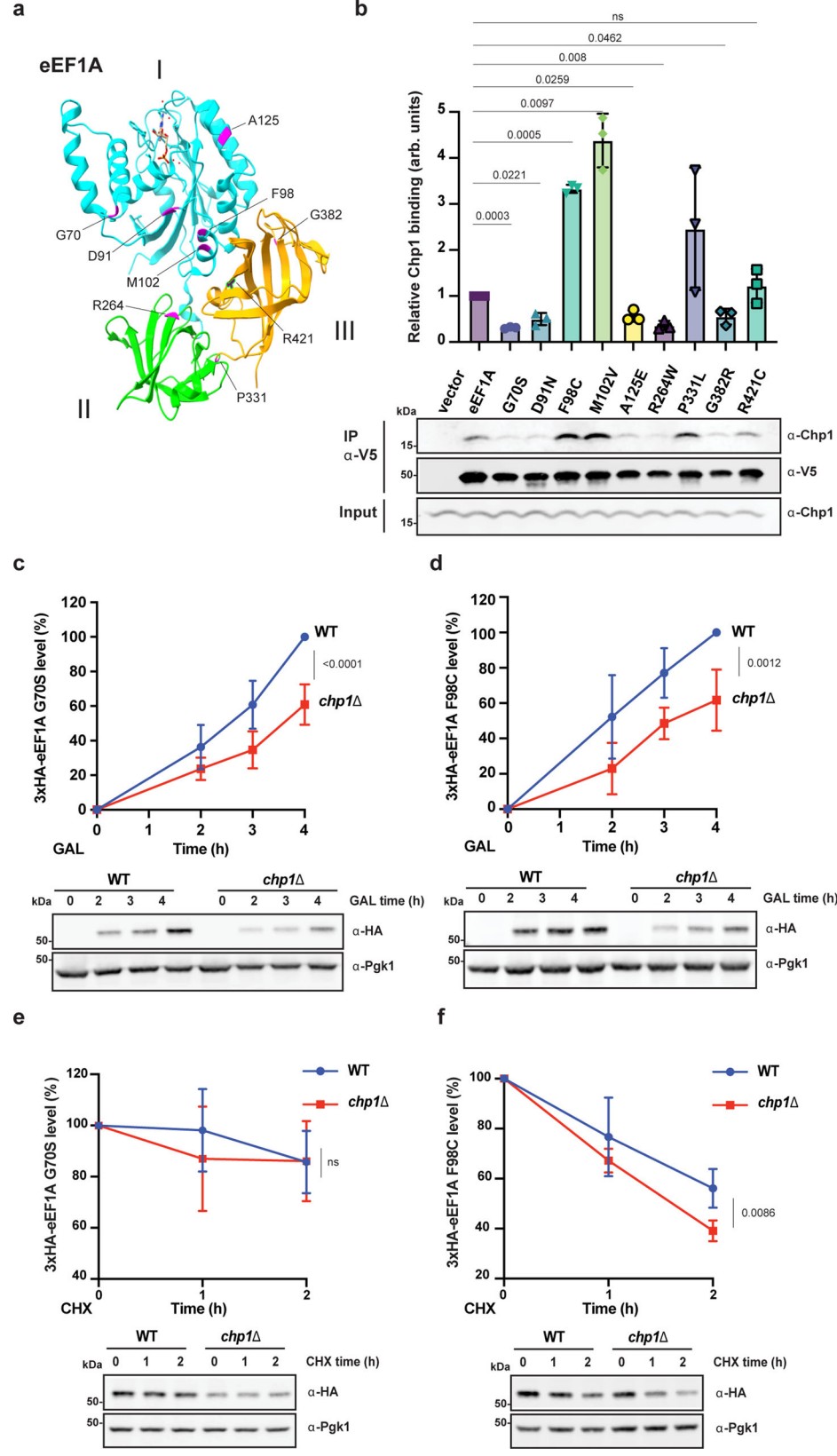

**Fig. 5 | Chp1 binds and stabilizes fully-synthesized eEF1A F98C variant that is associated with epileptic-dyskinetic encephalopathy. a** The position of the human eEF1A2 pathogenic mutations (magenta) mapped on the 3D structural representation of eEF1A from yeast (PDB: 1G7C). Domains I, II and III of eEF1A are shown in cyan, green and orange respectively. **b** Co-IP of the eEF1A pathogenic variants in **a** (V5-eEF1A) with Chp1 (means ± SD, *n* = 3, one-way ANOVA). **c** De novo expression of the pathogenic variant G70S of eEF1A in WT and *chp1*Δ cells following

galactose (GAL) induction (means ±SD, *n* = 5, two-tailed *t* test). **d** As in (**c**) but the F98C variant was analyzed (means ±SD, *n* = 5, two-tailed *t* test). **e** Stability of the variant G70S of eEF1A in WT and *chp1*Δ cells following translation arrest with cycloheximide (CHX) 3 hours after galactose induction (means ± SD, *n* = 4, two-tailed *t*-test). **f** As in (**e**) but the F98C variant was analyzed (means ± SD, *n* = 4, two-tailed *t* test). For each experiment n= independent cultures.

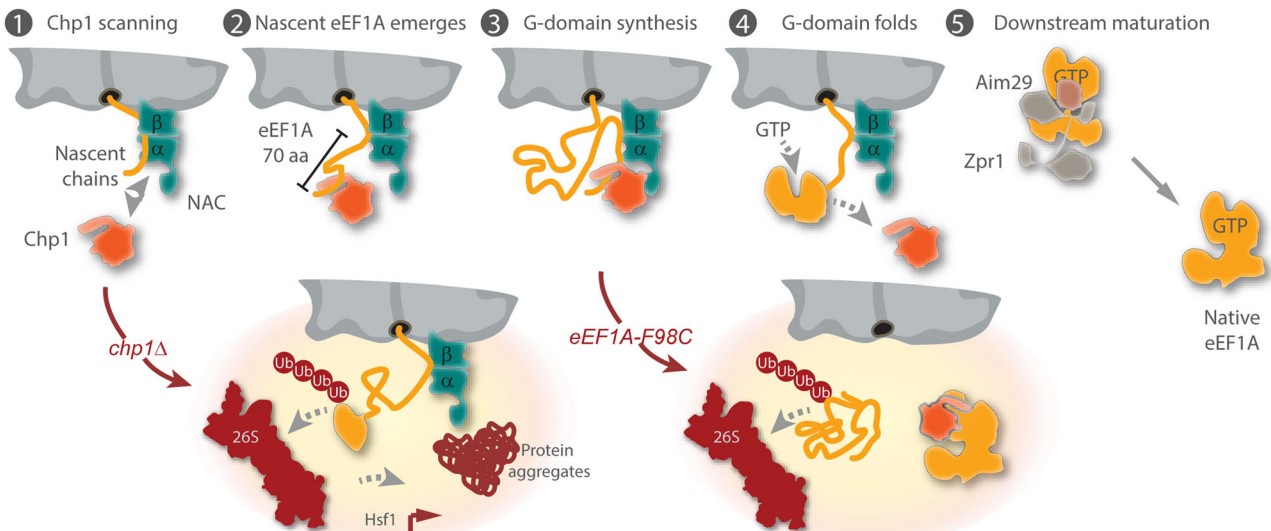

**Fig. 6 | Model for eEF1A biogenesis.** (1) Chp1 scans ribosome-nascent chain complexes. (2) Upon emergence of the first N-terminal 70 amino acids of nascent eEF1A (β1-α1-α2-α3) outside the ribosomal tunnel, Chp1 engages the nascent polypeptide by direct binding. (3) Chp1 remains bound to nascent eEF1A during the synthesis of the GTPase domain (domain I) and stabilizes it. Direct binding between Chp1 and the α-subunit of NAC through its UBA domain helps to stabilize Chp1 interaction with the ribosome- eEF1A domain I nascent chain complex. (4) Chp1 releases eEF1A nascent chain once the GTPase domain is completely translated, exposed out of the ribosomal tunnel and folded into its native stable conformation. (5) Following the expression of domains II and III, the Zpr1 chaperone and its cochaperone Aim29 mediate the downstream folding of the newly synthesized eEF1A into its final native conformation[17,18]. (Low-left inset:) Failure to recruit Chp1 (*chp1Δ*) during eEF1A biosynthesis impedes the folding of the GTPase domain and eEF1A is targeted to proteolysis by the 26S proteasome via polyubiquitylation. The high rates of aberrant eEF1A translation results in a heavy load on the proteostasis system; cells accumulate protein aggregates and activate Hsf1-mediated transcription that is responsive to the accumulation of misfolded proteins. (Low-right inset:) Chp1 binds fully synthesized eEF1A F98C which is an unstable variant of eEF1A associated to epileptic-dyskinetic encephalopathy in humans. Interaction of Chp1 with fully synthesized eEF1A F98C increases the stability of the mutant variant and protects it from proteosomal degradation.

WT or the G70S variant, and the turnover of the F98C variant was significantly accelerated in *chp1Δ* cells (Fig. 5e, f and Supplementary Fig. 8f). These results indicate that the F98C mutation located on a region of the GTPase domain important for nucleotide binding, severely impairs eEF1A stability which allows Chp1 to access its interaction site on the misfolded fully-synthesized protein. Chp1 interaction with the destabilized mutant protects it from removal by the UPS. In conclusion, Chp1 impacts on the biogenesis and also on the posttranslational quality control of eEF1A F98C mutant derived from patients with *EEF1A2*-linked cortical/cerebellar atrophy.

## Discussion

The multi-domain GTPase eEF1A is essential for translation and strong evolutionary selection has optimized its biosynthesis to meet the high-level cellular demands. The complex multi-domain architecture of eEF1A, the intrinsic metastability of GTPases and its high rate of synthesis most certainly impose a challenge for the general folding machinery, raising the question if a dedicated pathway safeguards its biosynthesis. Our data show that the conserved and previously uncharacterized protein Chp1 serves as a dedicated chaperone that collaborates with NAC at the ribosome to assist eEF1A biogenesis. Additionally, Zpr1-Aim29 has recently been shown to help in the folding of newly synthesized eEF1A[17,18]. Thus, eukaryotic cells have evolved a dedicated biogenesis pathway involving multiple components, which secures eEF1A folding. Importantly, this pathway starts early during eEF1A translation already at the ribosomal tunnel exit. Our working model for how these factors function in the biogenesis of eEF1A is summarized in Fig. 6.

We find that Chp1 interacts with ribosome-eEF1A nascent chain complexes during the synthesis of its GTPase domain. SeRP and biochemical analysis indicate that Chp1 selectively binds the nascent GTPase domain of eEF1A cotranslationally as soon as the first 70 amino acids of the protein have emerged at the polypeptide tunnel exit. Chp1 uses its N-terminal α-helix to bind the β1-α1-α2-α3 of eEF1A in a specific conformation. Coexpression of the isolated GTPase domain and Chp1 in a heterologous *E. coli* system suggests that Chp1 interaction with the domain increases its stability and potentially protects it from degradation. Exposure of the complete GTPase domain at the ribosomal tunnel exit triggers the dissociation of Chp1, likely due to cotranslational folding of the domain into its stable nucleotide-bound structure. In support, Chp1 does not bind nucleotide-loaded native eEF1A but will engage the protein under experimental conditions that unfold it, including nucleotide depletion by EDTA and destabilization of the GTPase domain by mutations in the Switch II helix. The transient association of Chp1 with specifically the nascent GTPase domain suggests that the chaperone must secure the commitment to a productive folding path of this domain prior to the expression of the beta-sheet rich domains II and III. Later on, when all three domains have been synthesized, Zpr1-Aim29 play a role in the final maturation of eEF1A. Interestingly, structural prediction as well as biochemical data indicate that Chp1 as well as Zpr1-Aim29 binds the α1-α2-α3 motif of the GTPase domain, suggesting that the unprotected structure may be vulnerable to off-pathway interactions, perhaps intramolecular interactions. Thus, it is likely that the arrangement of eEF1A with an N-terminal GTPase domain followed by the beta-sheet rich domains II and III presents a folding challenge during the synthesis of the translation factor, which is solved by the sequential engagement of dedicated chaperones.

The ribosome-associated chaperone NAC forms a heterotrimeric complex with Chp1 and its removal results in premature disengagement of Chp1 from the nascent GTPase domain and overall decreased interactions of Chp1 with ribosomes. Our data suggest that NAC plays a supportive role in eEF1A biogenesis perhaps by directly stabilizing the interaction of Chp1 with nascent eEF1A thereby increasing the local concentration of Chp1 at the ribosomal tunnel exit. Nevertheless,

inactivation of NAC does not result in decreased eEF1A levels raising the possibility that Chp1-NAC interaction does not directly facilitate eEF1A biogenesis. Recent findings regarding the SRP recruitment show that the α-NAC UBA domain transiently captures SRP to permit scanning of nascent chains and handing over the signal sequence to this targeting factor[11]. We find that also Chp1 binds α-NAC dependent on the UBA domain, suggesting that the NAC UBA domain may act as a coordinating factor of diverse nascent chain interactions.

In cells lacking Chp1 the expression of eEF1A is impeded, and newly synthesized eEF1A is rapidly targeted for UPS-mediated degradation. Such aberrant and futile synthesis of the highly expressed eEF1A burdens the proteostasis system, perhaps by saturating the limiting UPS capacity. As an outcome, cellular eEF1A protein levels are decreased, protein aggregates accumulate, the stress transcription factor Hsf1 is activated and cells become stress sensitive. Yeast genetic analysis as well as RP show that these phenotypes are directly linked to the damage cause by the abundant production of misfolded eEF1A rather than to global translation defects. The phenotype is reminiscent of Zpr1 inactivation supporting their involvement in a common pathway. In line, we find that *CHP1* and *ZPR1* share a set of genetic interactions with translation-associated genes. Moreover, the *chp1Δ* phenotype regarding eEF1A expression becomes accentuated upon eEF1A overexpression, suggesting that downstream components, perhaps Zpr1-Aim29, fail to handle the increased synthesis rates.

The concept of dedicated chaperones at the ribosome may extend more generally to the folding of other abundant translation-associated GTPases. Curiously, the entire GTPase domain of *E. coli* EF-G remains unfolded until domain synthesis is completed and the biogenesis of its ortholog eEF-2 has recently been shown to depend on specialized chaperone Hgh1, although it is unclear if it acts at the ribosome[35–38]. Interestingly, our SeRP data does not lend support to the scenario that Chp1 interacts with other proteins structurally related to eEF1A, which let us hypothesize that the chaperone solutions for these ancient GTPases is highly specific. For abundant proteins that are produced at high rates but difficult to fold, such as GTPase translation factors, dedicated folding mechanisms at the ribosome may be a requisite to adapt the general chaperone machinery at the polypeptide tunnel exit suggesting that there are more dedicated biogenesis factors to identify.

The key chaperone role of Chp1 is further supported by analysis of destabilizing *EEF1A2* mutations identified in patients with cortical/cerebellar brain atrophy. In addition to safeguarding the biogenesis of the mutants, Chp1 also associates with and stabilizes fully-synthesized eEF1A F98C (switch II region of the GTPase domain), demonstrating a potential role also in post-translational protein quality control. Hence, Chp1 association modifies the quality control pathways of destabilized eEF1A, suggesting it is important to understand Chp1 function to approach the pathogenesis of *EEF1A2*-linked cortical/cerebellar brain atrophy. In extension, perhaps disease phenotypes may be suppressed by modifying Chp1 activity.

The highly abundant eEF1A executes an essential function in mRNA translation. The requirement for high production rates of eEF1A may not be compatible with the typical translation pausing and progressive folding of nascent polypeptide segments once they emerge from the ribosomal tunnel exit. Hence, dedicated cotranslational folding factors are required to safeguard its biosynthesis. Misfolding of eEF1A is associated with extensive collateral damage on the proteostasis system with a negative impact on cellular fitness. Our findings demonstrate that the dedicated ribosome-associated chaperone Chp1 safeguards on-pathway folding of nascent eEF1A and thus, secures ample production of this essential translation factor and protects the proteostasis system from a significant protein misfolding burden.

## Methods

### Yeast strains and plasmids

Yeast strains and plasmids used in this study are listed in Supplementary Tables 1 and 2, respectively. The strains were grown in yeast-peptone-dextrose (YPD) medium or synthetic complete (SC) medium to select for strains transformed with plasmids. For the galactose-induced expression, strains were grown in YP supplemented with 2% raffinose over-night and diluted in YP supplemented with 2% galactose (YPGal). All strains are derivatives of the BY4743[39]. CAY1366 and CAY1367 are transformants of CAY1015 and a *HSP104*-eGFP BY4741-derivative, respectively obtained using a *chp1Δ::kanMX* PCR product. CAY1371 was obtained by transforming CAY1015 with a sfGFP-kanMX PCR product. JQY5 is a CAY1015 transformant obtained by integrating 4xFLAG-kanMX after *CHP1*. JQY7, JQY9, MMY139 and MMY141 were obtained by transforming CAY1366, CAY1015, MMY130 and MMY131 respectively with the PstI-restricted yeast integrating plasmid (YIp) pJQ24. JQY10 and JQY11 are Ura$^+$ transformants of CAY1015 and CAY1363, respectively and were obtained by replacing 440 bp of the endogenous promixal *TEF2* promoter by integrating a 1.8 kb *URA3*-$P_{TDH3}$-6xHis PCR fragment. MMY09 was constructed by transforming CAY1015 with two PCR products, containing $P_{CHP1}$ and sfGFP-kanMX respectively, to delete *CAN1*. Gene deletion and endogenous tagging in AMY58, MMY12, MMY13, MMY14, MMY17, MMY18, MMY24, MMY92, MMY94, MMY111, MMY105, MMY108, MMY130, MMY131, MMY132, MMY134, MMY149 have been performed by standard methods[40]. *nacΔ* strains (MMY66, MMY69 and MMY89) were obtained by transforming CAY1015, CAY1366 and CAY1371 with natNT2, hphNT1 and hisMX6 cassettes to delete *EGD2*, *BTT1* and *EGD1*, respectively[40]. MMY150, MMY152, MMY154, MMY155, MMY157 and MMY158 were constructed by transforming CAY1015 and CAY1366 with two PCR products, containing $P_{GAL1}$-HA-NatNT2 and *TEF2* WT, G70S or F98C, for integration into *LYS2*.

### Protein expression and purification

The proteins Chp1-Myc-6xHis, Chp1(Q18-Bpa)-Myc-6xHis, Chp1(V21-Bpa)-Myc-6xHis and Chp1(E25-Bpa)-Myc-6xHis were expressed from a T7 promoter in *E. coli* BL21-SI/pCodonPlus strain. Chp1-mCherry, mCherry, and eEF1A domain I were also expressed from the T7 promoter in *E. coli* BL21-SI/pCodonPlus cells but with N-terminal cleavable 6xHis-SUMO tags as previously described[41]. For co-expression of 6xHis-SUMO-Chp1 with Egd1 and Egd2 in *E. coli*, polycistronic variant plasmids with T7 promoters were used (pCA1039; pCA1038; pMM14). Dicistronic plasmids encoding 6xHis-SUMO-Chp1 together with either full-length eEF1A (pJQ12), domain I of eEF1A (pJQ15) or N-terminal amino acids 1 to 70 of eEF1A (pJQ20) were used for co-expression of these proteins in *E. coli*. The dicistronic plasmids pJQ32, pJQ33 and pJQ34 were used for coexpression in *E. coli* of domain I of eEF1A with 6xHis-SUMO-Chp1$_{10A}$ (T3A, F4A, E7A, T8A, L12A, D14A, I15A, F19A, V21A, E25A), 6xHis-SUMO-Chp1$_{6A}$ (L12A, D14A, I15A, F19A, V21A, E25) or 6xHis-SUMO-Chp1$_{Δ2-28}$, respectively. Proteins expression was induced at $OD_{600} = 1$ by the addition of 0.2 M NaCl and 0.5 mM IPTG in 2xYTON supplemented with appropriate antibiotics and when applicable 1 mM Bpa and pEVOL-pBpF were used. Following culturing at 30 °C for 2-3 h, cells were harvested and disrupted in LWB150 buffer (40 mM Hepes-KOH pH 7.4, 150 mM KCl, 5 mM $MgCl_2$, 5% (v/v) glycerol,) freshly supplemented with 1 mM phenylmethylsulfonyl fluoride (PMSF) and a few crystals of DNAse I using an EmulsiFlex-C3 (Avestin, Ottowa, Ontario, Canada). The cell-free protein lysates (centrifugation at 27,000 × g for 30 minutes) were subjected to IMAC purification with Macherey-Nagel™ Protino™ Ni-IDA (Thermo Fisher Scientific, Waltham, Massachusetts, USA). Protein was eluted with LWB150 supplemented with 200 mM imidazole. Proteins with 6xHis-SUMO tag were digested with Ulp1 protease to remove the tag while dialyzing against LWB150 buffer. The protein samples were depleted

from the tag by incubation with Ni-IDA resin. Proteins were finally dialyzed against LWB150 buffer for storage at −80 °C.

6xHis-eEF1A was purified from the yeast strain JQY10. Yeast cells were cultured on yeast-peptone (YP) supplemented with 4% glucose at 30 °C with shaking to $OD_{600}$ of 3-4. Cells were harvested by centrifugation and lysed in LWB150 freshly supplemented with 1 mM PMSF and cOmplete™, EDTA-free Protease Inhibitor Cocktail (Roche, Basel, Switzerland) using an EmulsiFlex-C3 high-pressure homogenizer at 25,000 PSI. The cleared cell lysate (centrifugation at $27,000 \times g$ for 30 minutes) was subjected to affinity purification using Ni-IDA resin and protein was eluted with LWB150 supplemented with 200 mM imidazole. Following removal of imidazole via dialysis against LWB150 buffer, the sample was diluted 3-fold in LWB0 (LWB150 but without KCl) to decrease the KCl concentration to 50 mM. The solution was slowly passed over a gravity column with DEAE-Sephacel resin (Cytiva Sweden AB, Uppsala, Sweden) previously calibrated with LWB50 (LWB150 but 50 mM KCl) to absorb contaminants. The KCl concentration was adjusted to 150 mM and protein was stored at −80 °C.

## Polysome profiling
Yeast strain JQY5, that expresses Chp1-4xFLAG, was grown in YPD to 0.5 $OD_{600}$. Cells were harvested via centrifugation at $3,000 \times g$ for 5 minutes at room temperature, placed on ice and washed with 5 mL of ice cold HEPES: Polymix buffer (20 mM HEPES:KOH pH 7.5, 2 mM dithiothreitol (DTT), 10 mM Mg(OAc)$_2$, 95 mM KCl, 5 mM NH$_4$Cl, 0.5 mM CaCl$_2$, 8 mM putrescine, 1 mM spermidine)[42] supplemented with 100 µg/mL cycloheximide, with or without 1 mM puromycin. Cells were pelleted again at 4 °C, resuspended in 250 µL of the same buffer containing 1 mM DTT and 1 × EDTA-free protease inhibitor cocktail tablet and then lysed with 0.25 g of 0.55 mm glass beads using a FastPrep-24 with two 20 s cycles at 4 m/s. Lysates were cleared by centrifugation at $10,000 \times g$ for 30 min at 4 °C. 5 Abs$_{260}$ OD units were loaded onto 12 mL 7−45% sucrose gradients, equilibrated with HEPES: Polymix buffer supplemented with 1 mM DTT, 100 µg/mL cycloheximide +/− 1 mM puromycin. Gradients were resolved by centrifugation at $209,500 \times g$ for 3 hours at 4 °C in a Beckman SW41 Ti rotor. Gradients were subsequently analyzed by continuous measurement of absorbance at 260 nm and fractionated using a Gradient fractionator (Biocomp instruments). Fractions were separated by SDS-PAGE and analyzed by immunoblotting using anti-FLAG M2 mouse 1:1000. Membranes were subsequently incubated with HRP-conjugated secondary antibodies (anti-mouse HRP; Rockland) at 1:10000 dilution. Visualization was performed on ImageQuant LAS 4000 (GE Healthcare) imaging system using Pierce® ECL Western blotting substrate (Thermo Scientific).

## Ribosome Profiling sample preparation (Total Translatomes)
Yeast cells were grown in 1 L YPD media at 30 °C and harvested at an $OD_{600}$ of 0.4-0.5. Cells were frozen with liquid nitrogen and pulverized together with frozen lysis buffer (20 mM HEPES pH 7.5, 150 mM KCl, 20 mM MgCl$_2$, 0.1 mg/mL cycloheximide, 0.1% (vol/vol) NP-40, 1 mM PMSF, EDTA-free protease inhibitor tablet, 0.02 U/µL DNase I) by mixer milling (2 min, 30 Hz, MM400 Retsch). Cell lysates were thawed by incubation at 30 °C for 2 min and cleared by centrifugation ($20,000 g$, 2 min at 4 °C). Samples were digested with 10 U /A$_{260}$ of total RNA with RNase I (Ambion) for 30 min at 4 °C. Ribosomes were isolated using sucrose cushions. 800 µL of sucrose cushion buffer (20 mM HEPES pH 7.5, 150 mM KCl, 20 mM MgCl$_2$, 0.1 mg/mL cycloheximide, 25% sucrose, EDTA-free protease inhibitor tablet (Roche)) was overlaid with lysate containing 100−200 µg of total RNA and centrifuged for 90 min at $245,000 \times g$ (TLA120 rotor, Optima Max ultracentrifuge 130,000 rpm, Beckman Coulter, Inc., Brea, CA) at 4 °C.

## Selective ribosome profiling (SeRP) sample preparation
Yeast cells were collected and lysed as for the general ribosome profiling. GFP binder coupled Sepharose beads[26] were washed by adding 2 times bead volume of wash buffer (20 mM HEPES, pH 7.5, 150 mM KCl, 20 mM MgCl$_2$, 0.1 mg/mL cycloheximide, 0.1% (vol/vol) NP-40, 10% Glycerol, 1 mM PMSF, EDTA-free protease inhibitor tablet, 0.02 U/µL DNase I) and centrifuging the beads at $450 \times g$ for 1 min at 4 °C to remove and discard the supernatant. This procedure was repeated 3 times. Cell lysates were thawed by incubation at 30 °C for 2 min and cleared by centrifugation ($20,000 \times g$, 2 min at 4 °C). Total RNA was quantified and 200 µg of the sample was used to prepare a total translatome sample as described above. 2700 µg of RNA was used to prepare the selective-translatome sample. Chp1 affinity purification and RNase digestion were performed simultaneously by incubating the sample with 250 µL of GFP binder coupled Sepharose beads and with RNase I (10 U /A$_{260}$ of total RNA) for 30 min at 4 °C. Samples were thoroughly washed, once with 1 mL lysis buffer and 3 times with 1 mL wash buffer, for 5 min, rotating at 4 °C. After each washing step the buffer was exchanged by sedimenting the beads ($450 \times g$, 1 min at 4 °C).

## Differential expression and gene set enrichment analysis
Ribosome profiling results reporting on total translatomes of WT and chp1Δ cells were used to carry out differential expression (DE) analysis with the Bioconductor package DeSeq2 (v1.28.1)[43]. Count data from two biological replicates for each strain were used to calculate log2 fold changes, shrunken log2 fold changes (using the 'apeglm' algorithm)[44], p-values and adjusted p-values with this package.

The Bioconductor package ClusterProfiler (v3.16.1)[45] and DOSE (v3.14.0)[46] were used to carry out Gene Set Enrichment Analysis (GSEA) following the DE analysis. Genes were ranked in a decreasing order according to the shrunken log2 fold change values calculated with DeSeq2 and tested for enrichment of gene sets among higher or lower rank positions. Benjamini & Hochberg (BH) method was used to adjust p-values for multiple comparisons.

## Ribosome profiling library preparation
All ribosome profiling and selective ribosome profiling libraries were prepared as described previously[26]. Depletion of most prevalent rRNA fragments were carried out using biotinylated reverse-complement DNA oligonucleotides (developed in collaboration with siTOOLs Biotech) as described previously[47].

## Sequencing and data processing of RP data
The cDNA libraries were sequenced on a NextSeq550 (Illumina) according to the manufacturers protocol. 3' Adapters were trimmed using Cutadapt v 3.2 with the following command: cutadapt --cores=8 --nextseq-trim=20 -m23 --discard-untrimmed -O6 -a ATCGTAGATCGGAAGAGCACACGTCTGAACTCCAGTCAC -o '<output_path>/'outfile.fastq.gz' '<input_path>/infile.fastq.gz' 1> '<output_path>/'Cutadapt_report.txt. Unique molecular identifiers (UMI) were removed using a custom Julia script (Script 1). julia -p 8 '<script_path>.jl' '<input_path>/'infile.fastq.gz' '<input_path>/outfile.fastq.gz' --umi3 5 --umi5 2. To remove rRNA, the reads were aligned to S. cerevisiae rRNA sequences with bowtie2 v 2.4.2 and only the reads that do not align were kept: bowtie2 -p8 -t -x '<ref_file_path>' -q infile.fastq.gz' --un '<outfile_path>/outfile.fastq' -S /dev/null 2> '<outfile_path>/Bowtie2_report.txt'. The remaining rRNA depleted reads were aligned to the S. cerevisiae S288C Genome (Assembly: GCA_000146045.2) with STAR v 2.7.7a. The TEF2 entry in the fasta file was removed to allow both TEF1 and TEF2 reads to align to the TEF1 sequence. The STAR index with the modified fasta file was generated using the following command: STAR --runMode genomeGenerate --runThreadN 8 --genomeDir <input_path>--genomeFastaFiles infile.dna.toplevel.fa --genomeSAindexNbases 10 --sjdbGTFfile <out-file_path>/outfile.gtf. As TEF1 and TEF2 differ from each other by two mutations at position 762 (T->C), and position 834 (T->C), the threshold for the allowed maximum number of mismatches per pair

was set to 2. The alignment was performed with the following command: STAR --runThreadN 7 --genomeDir '<ref_file_path>' --readFilesIn '<infile_path>/infile.fastq' --outSAMmultNmax 1 --outFilterType BySJout --outFilterMismatchNmax 2 --alignIntronMin 5 --outFileNamePrefix '<outfile_path>' --outReadsUnmapped Fastx --outSAMtype BAM SortedByCoordinate --outSAMattributes All XS --quantMode GeneCounts --twopassMode Basic --limitBAMsortRAM 1185598524. Reads from PCR duplicates were collapsed to a single read and positions of ribosomal a, p, and e-sites on footprints were assigned with a Julia script (Script 2): julia -p 8 <script_path>script.jl -g <annotation_file_path>/annotation_file.gff3 -o <out-put_path>/<input_path>/infile.bam" -u -c 1. The data was analyzed using a custom R package RiboSeqTools, accessible from https://github.com/ilia-kats/RiboSeqTools[48]. The analysis was performed exclusively using p-site assigned reads for all datasets.

### In vitro Chp1- NAC and Chp1- eEF1A interaction assay

Purified Chp1-Myc-6xHis and NAC or NAC (ΔUBA) were mixed at 10 μM and Chp1-Myc-6xHis and eEF1A domain I were mixed at 4 μM in a reaction volume of 170 μL in LWB150 supplemented with 20 mM imidazole. The samples were incubated for 1 h with slow rotation at room temperature. Following the addition of 50 μL of Ni-NTA Magnetic Agarose Beads (50% slurry) (Qiagen) and 30 min continued incubation, the matrix was washed 3 times with 200 μL LWB150 buffer + 20 mM imidazole. Bound protein was eluted by the addition of 50 μL of LWB150 buffer supplemented with 250 mM imidazole. Samples were analyzed by western blot and Coomassie Brilliant Blue staining.

Purified Chp1-mCherry and 6xHis-eEF1A were mixed at 2 μM in a reaction volume of 170 μL in LWB150 at 4 °C. Samples were incubated for 1 h with slow rotation. Following addition of 10 μL slurry of RFP-Trap® Magnetic Agarose (Chromotek, Planegg, Germany) and 1 h continued incubation, the matrix was washed 3 times with 200 μL LWB150 buffer and bound protein was eluted by the addition of 50 μL of 2 × SDS sample buffer. To assess the effect of nucleotide binding, 6xHis-eEF1A was pre-incubated for 20 min with 1 mM of GMP-PNP or with 10 mM of EDTA. To assess the effect of stabilization by glycerol, 6xHis-eEF1A was pre-incubated with glycerol 25% (v/v) alone or in combination with 1 mM GMP-PNP or 10 mM EDTA. Following SDS-Page the Coomassie Brilliant Blue signals corresponding to 6xHis-eEF1A and Chp1-mCherry in the bound fractions were quantified using the Image Lab software after image acquisition with ChemiDoc™ XRS+ (BioRad). The signals were normalized to protein molecular weight and the normalized ratios 6xHis-eEF1A/Chp1-mCherry (in percentage) were plotted for each tested condition.

### Size-exclusion chromatography

Purified Chp1-eEF1A variant complexes (250-400 μg) were loaded onto a Superdex 200 increase 10/300 GL column (GE Healthcare) at a flow rate of 0.4 mL/min in LWB150 buffer. Eluted fractions corresponding to peaks of OD280 were collected and analyzed by SDS-PAGE.

### Bioluminescent assay for determination of Hsf1 activity

A yNlucPEST reporter was used to monitor Hsf1 activity as previously described[49]. Briefly, Nano-Glo substrate (Promega, Madison, WI, USA) was diluted 1:100 with the supplied lysis buffer and mixed 1:10 with logarithmic growing cells carrying the reporter plasmid pCA955. Bioluminescence was determined after 3 min of incubation, using an Orion II Microplate Luminometer (Berthold Technologies GmbH, Bad Wildbad, Germany) and the obtained signal was normalized to $OD_{600}$.

### Protein extraction, SDS-PAGE and immunoblot analysis

Protein extracts were prepared from cells in logarithmic phase grown in YPD or selective media. Briefly, culture aliquots were incubated on ice for 10 min after addition of precooled NaOH to a final concentration of 0.37 M. Trichloroacetic acid was then added to a final

concentration of 8.3% (w/v). The next day, cells were collected by centrifugation and the pellets were rinsed with 1 M Tris base. Samples were re-suspended in 4% SDS sample buffer and boiled. Equal amounts of SDS-solubilized protein were separated by SDS-PAGE, transferred to Amersham Protran Supported 0.45 mm Nitrocellulose Blotting membrane (GE Healthcare, Chicago, IL, USA) and incubated at room temperature with the primary antibody: α-eEF1A rabbit 1:10000 (ED7001; Kerafast, inc.), α-Fes1 rabbit 1:5000[50], α-Btn2 rabbit 1:5000[51], α-Pgk1 22C5D8 mouse 1:5000 (459250; Thermo Fisher Scientific), anti-NAC 1:5000 rabbit;[8] anti-HA 3F10 Rat 1:5000 (11867423001 Roche product line, Merck KgaA), anti-GFP 7.1/13.1 mouse 1:5000 (11814460001, Roche product line, Merck KgaA), anti-Chp1 1:1000 rabbit (this study), anti-V5 Sv5-Pk1 mouse 1:5000 (R960-25; Thermo Fisher Scientific), anti-GAPDH 1D4 mouse 1:5000 (MA1-16757; Thermo Fisher Scientific), anti-ubiquitin HRP P4D1 mouse 1:1000 (sc-526508; Santa Cruz Biotechnology, Inc), anti-6X His tag HIS.H8 mouse 1:5000 (ab18184; Abcam plc.), anti-Rpl8 (recognizes yeast Rpl2) rabbit 1:1000 (PA5-41713; Thermo Fisher Scientific), anti-Rpl25 rabbit 1:5000[52], anti-Egd2 rabbit 1:2000 (this study), anti-Hsp42 rabbit 1:5000[53], anti-FLAG M2 mouse 1:10000 (F1804; Merck KGaA), anti-Myc peroxidase 9E10 1:5000 (11814150001; Roche product line, Merck KgaA). After incubation with secondary antibodies (Li-COR Biosciences, Lincoln, NE, U.S.A.) the specific protein signal was detected using the Odyssey Fc infrared imaging system (Li-COR Biosciences). Signal quantification was performed using the Image Studio 3.1.4 (LI-COR Biosciences) by normalizing the background-corrected signals to the loading control.

### Growth assays

For drop plates, yeast cells were grown overnight in liquid media at 30 °C with shaking, diluted next morning to $OD_{600}$ 0.15-0.2 and allowed to grow in the same conditions to logarithmic phase before serial 1:10 dilutions and spotting onto solid medium. Plates were incubated at 30 °C and 39 °C for 3 days. For liquid growth assays, cells were incubated overnight in SC medium and were inoculated to $OD_{600}$ 0.1 in 200 μL of the indicated medium in 96-well microplates with clear, flat bottom. Optical density was assessed every hour for 22 h with a 2300 EnSpire™ plate reader (Perkin Elmer, Waltham, MA, USA). Doubling times (in hours), were calculated as $([(T2-T1) *\ln(2)])/([LN(D2)-LN(D1)])$, T1: time 1, T2: time 1+4 h, D1: $OD_{600}$ at T1, D2: OD600 at T2. All the calculations were made using T1 and T2 in the exponential growth phase.

### Fluorescence microscopy

Cells corresponding to 1 mL $OD_{600}$ 1 (1 $OD_{600}$ unit) were harvested by centrifugation (1 min, 3500 × g), washed with PBS and fixed with 4% paraformaldehyde solution for 10 min on ice. After washing and resuspension of the cells in PBS, confocal microscopy pictures were taken with a ZEISS LSM800 Airyscan microscope using ZEISS ZEN software control. Plan-Apochromat 63x/1.40 Oil M27 objective was used with appropriate filter settings to visualize GFP. Micrographs shown are Z-projections obtained by using Maximun intensity projection method and they were analyzed and processed with the open-source software Fiji[54].

### Co-immunoprecipitation

Cells (50 $OD_{600}$ units) were harvested and lysed on ice in 300 μL lysis buffer (20 mM Tris pH 7.4, 150 mM NaCl, 2 mM $MgCl_2$, 2 mM EDTA, 5% glycerol, 0.1 g/L cycloheximide, 1 × cOmplete™, EDTA-free Protease Inhibitor Cocktail). The resuspended cells were transferred to a 1.5 mL screw-capped micro tube containing an equal volume of 0.4−0.6 mm diameter glass beads and lysed by bead beating. Unbroken cells and debris were removed by 5 minutes centrifugation at 2000 × g and the supernatants were transferred to fresh tubes and 0.25% NP-40 was added. Following incubation for 10 min on ice, the samples were centrifuged at 16,200 × g for 15 min at 4 °C

and 40 μL of each supernatant were collected as input samples. Supernatants were transferred to fresh tubes and the volumes were adjusted to 1 mL with lysis buffer before equilibrated GFP-Trap®/V5-Trap® (Chromotek) was added. Samples were incubated on a rotating wheel for 2 h at 4 °C and the beads were washed 5 times with lysis buffer before protein was eluted by incubation in 40 μL 4% SDS sample buffer for 10 min at 95 °C. Proteins were separated by SDS-PAGE and analyzed by immunoblotting.

## Pelleting yeast ribosomes
Cells (100 $OD_{600}$ units) were harvested and lysed by glass bead beating in 350 μL ice-cold lysis buffer (20 mM HEPES-KOH pH 7.4, 50 or 500 mM KOAc, 5 mM MgOAc, 4 mM DTT, 0.1 mg/mL cyclohex-imide, 1× cOmplete™, EDTA-free Protease Inhibitor Cocktail). Unbroken cells and debris were pelleted by 5 minutes of centrifugation at 2000 × $g$. The supernatants were transferred to a fresh tube and the protein lysates were diluted to 4 mg/mL (Bradford assay). 1% (w/v) CHAPS was added and the samples were incubated 5 minutes on ice before removing debris by centrifugation at 20,000 × $g$ for 20 min. 50 μL of each supernatant were saved and 300 μL of the lysates were transferred to a 13 mm × 51 mm poly-carbonate tubes with 900 μL 1 M sucrose cushions. Ribosomes were sedimented by centrifugation in an Optima™ MAX 130,000 Ultra-centrifuge with a TLA100.3 rotor (Beckman Coulter Inc. Brea, CA, USA) at 264,400 × $g$, 4 °C for 4 h. The supernatants (soluble fraction) were gently pipetted out of the tube and 50 μL of the soluble fractions were transferred in a fresh tube with 4% SDS sample buffer. The ribosome-enriched pellets were resuspended in 100 μL of lysis buffer by extensive manual stirring. The ribosomes pellet fractions were transfer in a new tube with 4% SDS sample buffer. All samples were boiled for 5 minutes at 95 °C. Proteins were separated by SDS-PAGE gels and analyzed by immunoblotting.

## Ubiquitylation assay
For precipitation of ubiquitinylated proteins under denaturing conditions, a yeast culture of 70 $OD_{600}$ units was harvested after treatment with 10 mM NEM and cells were lysed on ice using 2 mL 1.91 M NaOH, 7.5% v/v β-mercaptoethanol followed by the addition of 2 mL 55% tri-chloroacetic acid. After 15 min on ice, the samples were centrifuged 15 min at 16,000 × $g$ and the pellets were washed two times with 2 mL ice-cold acetone and were resuspended in 1.5 mL buffer A (6 M gua-nidium chloride, 100 mM $NaH_2PO_4$, 10 mM Tris-HCl pH 8.0, 0.05% Tween 20). 10 mM imidazole was added together with 100 μL Ni-NTA magnetic agarose beads (Qiagen, Hilden, Germany) and the samples were incubated on a rotating wheel for 16 h at 4 °C. The beads were washed two times with buffer A containing 20 mM imidazole and two times with buffer B (8 M urea, 100 mM $NaH_2PO_4$, 10 mM Tris-HCl pH 6.3, 0.05% Tween 20). The proteins were eluted by incubating the beads in 50 μL 4% SDS sample buffer for 10 min at 95 °C. Proteins were separated by SDS-PAGE and analyzed by immunoblotting.

## Isolation of protein aggregates
Protein aggregates from yeast cells incubated with 10 mM NEM for 10 minutes were isolated by centrifugation as a detergent insoluble material[55]. Yeast cells (500 $OD_{600}$ units) were harvested and lysed using a EmulsiFlex-C3 (Avestin, Ottowa, Ontario, Canada) in 10 mL ice cold lysis buffer (100 mM Tris-HCl pH 7.5, 200 mM NaCl, 1 mM EDTA, 1 mM DTT, 5% glycerol, 1 mM PMSF, 10 mM NEM, 1× cOmplete™, EDTA-free Protease Inhibitor Cocktail). Unbroken cells and debris were pelleted by 5 minutes centrifugation at 3000 × $g$ and discarded. Protein concentration was adjusted to 1.5 mg/mL (Bradford assay) and 1 mL of lysate was centrifuged at 20,000 × $g$ for 15 min. The super-natant (soluble fraction) was gently pipetted out of the tube and 50 μl of the soluble fraction was transferred in a fresh tube with 4% SDS sample buffer. The pellet was resuspended in 400 μL lysis buffer

supplemented with 2% NP-40 and sonicated three times for 5 minutes. Protein aggregates were sedimented by centrifugation at 20,000 × $g$ for 15 min. The pellet was resuspended in 80 μL 4% SDS sample buffer. All samples were boiled for 5 minutes at 95 °C. Proteins were separated by SDS-PAGE and analyzed by immunoblotting.

## In vitro photo-crosslinking
Purified Chp1-Myc-6xHis with p-benzoyl-L-phenylalanine (Bpa) incor-porated at position 18, 21 or 25 was mixed with eEF1A domain I or mCherry negative control at 8 μM in a reaction volume of 80 μL in LWB150 buffer. Samples were transfered to a 96-well plate on ice and exposed to UV for 10 min using a Sylvania CF-S 9W/BL350 fluorescent lamp. Samples were analysed by SDS-PAGE followed by Coomassie Brilliant Blue staining. Similarly, 6xHis-eEF1A (pre-incubated with either EDTA (10 mM), GMP-PNP (1 mM) or glycerol (25% v/v)) was mixed with Chp1-Myc-6xHis with Bpa incorporated at position 25.

## In vivo photo-crosslinking
Cells carrying pMM09 or pMM10 encoding Chp1-HA-myc-8xHis with an amber mutation introduced at codon 12 or 122 and ECRYS-BpA for p-benzoyl-L-phenylalanine incorporation[50,55,56] were grown at 30 °C to mid-log phase in selective media in the presence of 1 mM p-benzoyl-L-phenylalanine (Bachem) added from a 100 mM stock solution freshly prepared in 1 M NaOH. Cells were then resuspended in ice-cold water and irradiated with UV-A on ice using a Sylvania Lynx BL350 15 W fluorescent lamp for 1 h. Total protein samples were prepared by TCA precipitation and analyzed by western blotting.

## Two-step purification of Chp1-crosslinked proteins and MS analysis
Cells (600 $OD_{600}$ units) were harvested and lysed using a EmulsiFlex-C3 (Avestin, Ottowa, Ontario, Canada) in 20 mL ice cold lysis buffer (20 mM Tris-HCl pH 7.5, 150 mM NaCl, 1 mM PMSF, 2 mM EDTA, 2 mM $MgCl_2$, 1× Protease inhibitor complete, 5% glycerol). The cell-free protein lysates (centrifugation at 27,000 × $g$ for 30 minutes at 4 °C) were subjected to IMAC purification using Macherey-Nagel™ Protino™ Ni-IDA (Thermo Fisher Scientific, Waltham, Massachusetts, USA) fol-lowed by immunoprecipitation with Myc-Trap® (Chromotek). Eluted proteins were separated by SDS-PAGE gels and stained with Coomassie Brilliant Blue.

## LC-Orbitrap MS/MS
LC-Orbitrap MS/MS was performed at the Science for Life Laboratory of Uppsala, Sweden. The proteins were in-gel digested by trypsin according to a standard operating procedure. The collected peptides were purified with a ZipTip C18 (Millipore, MA) and then vacuum centrifuged to dryness. Thereafter the samples were dried and dis-solved in 30 μL 0.1% formic acid and further diluted 2 times. The resulting peptides were separated in reversed-phase on a C18-column and electrosprayed online to a QEx-Orbitrap mass spectrometer (Thermo Finnigan) with 90 min gradient. Tandem mass spectrometry was performed applying HCD. Database search was performed using the Sequest algorithm, embedded in Proteome Discoverer 1.4 (Thermo Fisher Scientific) against the database consisted of *Saccharomyces cerevisiae* proteome extracted from Uniprot, Release June 2018. The search parameters were set to Enzyme: Trypsin (Fixed modification was Carbamidomethyl (C), and variable modifications were Oxidation (M), Deamidated (NQ). The search criteria for protein identification were set to at least two matching peptides of 95% confidence level per protein.

## qPCR analysis
RNA was extracted from 15 $OD_{600}$ units of cells grown in YPD or YPGal using a RiboPure RNA Purification Kit for Yeast (Ambion, Invitrogen). cDNA was synthesized from DNase I-treated RNA using Superscript

III Reverse Transcriptase (Invitrogen) and qPCR was performed using KAPA SYBR Fast Universal qPCR Kit (KAPA Biosystems) with primers (TEF1/2 for 5´-TGGCTTTCACCTTGGGTGTT -3´; HA-TEF1 for 5´-GGGC TATCCCTATGACGTCC-3´; TEF1/2 rev 5´-CCCTTGTACCATGGAGC GTT- 3´; TAF10 for 5´-ATATTCCAGGATCAGGTCTTCCGTAGC-3´; TAF10 rev 5´-GTAGTC TTCTCATTCTGTTGATGTTGTTGTTG -3´). Quantification was performed using the 2−ΔΔCT method and expression was normalized to *TAF10*[55].

## ColabFold modeling and structural representations

ColabFold v1.5.2, AlphaFold2 using MMseqs2 was used for modeling of protein complexes using 5 models[57–59]. Parameters were set to model_type:auto; num_recycles:3; recycle_early_stop_tolerance:auto; relax_max_iterations:200; pairing_strategy:greedy; max_msa:auto; num_seeds:1; dpi:200. Models 1-5 (ranked by average plDDT values for the entire chain length), for the complexes Chp1:Egd2:Egd1 (Uniprot accessions; Q0897:P38879:Q02642), αNAC:βNAC:PBDC1 (Q13765:P20290:Q9BVG4) and eEF1A domain I:Chp1 (P02994 residues 1 to 250:Q0897) are supplied as Supplementary Data 2 and 3, respectively together with sequence coverage, predicted lDDT per position, predicted aligned error (PAE) and javascript object notation files (.json) for the runs. Structural representations were visualized using UCSF ChimeraX[60].

## Statistical analysis

Results are expressed as mean ± standard deviation. Experiments were carried out at least 3 times. Individual data points and hence number of biological replicates are represented in the figures. Statistical significance was analyzed with Graphpad Prism 7 with single T-test and multiple T-test (for pairwise comparison) or one-way ANOVA (Tukey's test for multiple comparisons).

## Reporting summary

Further information on research design is available in the Nature Portfolio Reporting Summary linked to this article.

## Data availability

The sequencing data generated in this study have been deposited in the NCBI's Gene Expression Omnibus[61] database under accession code GSE221651. The MS data generated in this study have been deposited to the ProteomeXchange Consortium via the PRIDE[62] partner repository under the accession code PXD043391. Source data are provided with this paper.

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

## Acknowledgements

We thank Sabine Rospert for sharing anti-Rpl25 antibodies, Elke Deuerling for anti-NAC antibodies, Verena Kohler for the Hsp104-GFP strain and Michael C. Kruer for p416GPD plasmids expressing V5-Tef2. We thank the MS Facility at Uppsala University for performing LC-Orbitrap MS/MS. This work was supported by the funds from Knut and Alice Wallenberg Foundation (2020-0037 to VH), ERC Advanced grant TransFold (743118) and ERC Synergy Grant CoTransComplex (101072047) to BB, European Regional Development Fund through the Centre of Excellence for Molecular Cell Technology (VH), grant PRG335 from the Estonian Research Council (VH), Heidelberg Biosciences International Graduate School, HBIGS (to IEK), Swedish Research Council (project grants 2019-04052 and 2023-04717 to CA and 2017-03783, 2021-01146, to VH); Swedish Cancer Society (20 1045 and 23 2949 to CA and 20 0872 to VH); Deutsche Forschungsgemeinschaft (DFG KR 3593/4-1 to GK).

## Author contributions

Conceptualization (MM, JQC, M Ciccareli, GK, BB, CA), Methodology (MM, JQC, KJ, KJT, M Ciccarelli, AEM, DL, VH, BB, CA), Software (IEK), Validation (MM, JQC, KJ, IEK, KJT, M Ciccarelli, AEM, DL, EG, M Czech, VH, CA), Formal analysis (MM, JQC, KJ, IEK, DL), Investigation (MM, JQC, KJ, IEK, KJT, M Ciccarelli, AEM, DL, EG, M Czech, VH, CA), Resources (MM, JQC, KJ, AEM, VH, BB, GK, CA), Data Curation (MM, JQC, KJ, IEK, DL), Writing - Original Draft (MM, JQC, CA) Writing - Review & Editing (MM, JQC, M Ciccarelli, BB, CA, VH, IEK, GK, DL), Visualization (MM, JQC, KJ, IEK, KJT, M Ciccarelli, CA), Supervision (BB, GK, CA), Project administration (VH, BB, GK, CA), Funding acquisition (VH, BB, GK, CA).

## Funding

## Competing interests

The authors declare no competing interests.
