## [Peer Review File · Nature Communications]

Chp1 is a dedicated chaperone at the ribosome that safeguards eEF1A biogenesisREVIEWER COMMENTS

Reviewer #1 (Remarks to the Author):

In this manuscript, Minoia et al. investigated a conserved and previously uncharacterized ribosome-associated factor (YPL225w), the absence of which triggers heat shock factor 1 (Hsf1) activation in yeast. The authors termed this factor chaperone 1 (CHP1) and found that it interacts with ribosomes that translate the protein eEF1A, a key translation elongation factor and one of the most abundant cellular proteins. Specifically, CHP1 binds to ribosomes when the N-terminal GTPase domain of eEF1A leaves the ribosomal tunnel suggesting that CHP1 helps to fold this domain during translation. The authors also provide evidence that knockout of CHP1 in yeast partially impairs eEF1A biogenesis and causes proteostasis stress. Additionally, the authors show an interaction between CHP1 and the N-terminus of eEF1A in a *E. coli* co-overexpression system. Based on these data, the authors conclude that CHP1 is a dedicated cotranslational chaperone for eEF1A.

Furthermore, the authors show that CHP1 interacts with the nascent polypeptide-associated complex (NAC), an abundant and ubiquitous heterodimeric complex located at the ribosome tunnel exit. NAC seems to stabilize ribosome binding of CHP1 through a direct interaction with the alpha-subunit of NAC. However, eEF1A biogenesis was not affected in the absence of NAC in yeast.

The discovery of a potential specific cotranslational chaperone for eEF1A that interacts with NAC is novel and would be of great interest to a wide readership. However, the presented data are too preliminary to provide a clear mechanistic model for the interaction of CHP1 with NAC and nascent eEF1A on ribosomes. For unknown reasons, obvious and simple experiments addressing the molecular interactions of CHP1 with eEF1A and NAC were not performed, and some conducted experiments appear to have technical flaws (see specific points below). The manuscript requires a major revision before I can support publication in a high-impact journal such as *Nat Commun*.

Major points:

1. The authors need to provide clear evidence that the alpha-subunit of NAC interacts with CHP1. Based on the author's ColabFold model (Fig. S1e), alphaNAC contacts CHP1 via the C-terminal UBA domain. For unknown reasons, this was not mentioned in the text and not addressed experimentally. The authors should perform interactions studies *in vitro* and *in vivo* with NAC variants lacking the UBA domain. In addition, they should also mutate the UBA binding interface on CHP1.
2. Related to the previous point, the photo-crosslinking analysis in Fig. S1c is questionable: the authors used an antibody that detects both NAC subunits (alpha and beta). How can the authors be sure that the crosslink is to alphaNAC and not betaNAC? There seems to be an additional, less prominent NAC crosslink above the marked band, which, considering the size difference between alpha- and betaNAC, could be a crosslink to alphaNAC, while the stronger, marked crosslink is a betaNAC crosslink. The authors need to clarify this.

3. The structure of the central heterodimerization domain of yeast NAC in the author's ColabFold model (Fig. S1e) is inconsistent with published structures of human NAC (PMID: 37347872, PMID: 35201867). The authors need to explain this. Is yeast NAC different or is just the model wrong? The authors should provide AlphaFold2 models of the NAC-CHP1 interaction of both, human and yeast, and compare the structures.

4. The authors demonstrate that the N-terminal domain of eEF1A is bound by CHP1. However, the authors did not experimentally investigate the substrate binding site of CHP1. According to the author's ColabFold model, a N-terminal helix of CHP1 interacts with eEF1A. This model needs to be tested by mutational approaches (deletion and point mutants of CHP1), both in vitro and in vivo.

5. For unknown reasons, the authors used a plethora of diverse affinity tags on CHP1 (GFP, mCherry, HA, FLAG, SUMO, etc), which appear to be located sometimes at the N-terminus and sometimes at the C-terminus. Considering the small size of CHP1 (~17 kDa), use of large tags like GFP and mCherry (both ~27 kDa) is dangerous and could result in artefacts. The cotranslational interaction studies in vivo was performed with a C-terminally tagged variant, CHP1-GFP. The authors should therefore stick to CHP1 variants with a C-terminal tag for their in vitro interaction studies. According to the authors model, a N-terminal helix of CHP1 interact with eEF1A. Therefore, a large tag like mCherry on the N-terminus should be avoided. I am in particular concerned with the analysis in Fig. S3f, where a mCherry tag was fused to the N-terminus of CHP1. Here, the authors wanted to prove that CHP1 only interacts with unfolded eEF1A, but not with natively folded eEF1A. The authors need to rule out that the mCherry tag blocks the substrate interaction.

6. The authors often use an E. coli protein co-overexpression system for protein-protein interaction studies. Why these artificial overexpression conditions in bacteria were used by the authors is puzzling. It would be better to express and purify the proteins individually and then perform interaction studies at physiological relevant protein concentrations in vitro in the test tube.

Minor points

1. The SEC analysis in Fig. S1d is questionable, because alphaNAC was expressed alone without betaNAC. This results in formation of artificial alphaNAC homodimers. These homodimers would have two CHP1 binding sites (e.g., two UBA domains). The authors should purify the heterodimeric NAC complex (with and without the UBA domain) and repeat this analysis.

2. The strongest evidence that eEF1A biogenesis is indeed impaired in CHP1 knockout cells is the analysis in Fig. 3f using the GAL-induced expression system (the differences in the eEF1A steady-state levels are very minor, and near the detection limit of immunoblotting ~30%). However, for the GAL system important controls are missing. The authors should induce expression of a control protein unrelated to CHP1 and/or provide mRNA levels of eEF1A.

3. The authors have done themselves no favors by analyzing a cotranslational NAC-related mechanism in yeast, the only known organism, in which NAC is not essential. There are clear functional differences between NAC in yeast and higher eukaryotes. If possible, the authors should perform PBDC1 and NAC knockdown experiments in human cells and analyze potential defects in eEF1A biogenesis (which might be much stronger than in yeast!).

Reviewer #2 (Remarks to the Author):

To identify ribosome-associated chaperones, Minoia et al. compared ribosome-associated proteins in cells with different gene deletions that activate Hsf1. They discovered previously unknown ORF YPL225W among the strongest activators of Hsf1. They name the gene CHP1 for Chaperone 1. Using Bpa crosslinking they find that Chp1 binds NAC subunit Egd2, and that NAC helps recruit Chp1 to ribosomes. Using selective ribosome profiling they identify TEF1/2 (encoding eEF1A) as the most enriched transcripts, which led them to suggest eEF1A is the sole nascent chain substrate of Chp1. Findings from various assays support their conclusions that NAC facilitates binding of Chp1 to the N-terminal GTPase domain of eEF1A, which stabilizes folding of eEF1A as it emerges from the ribosome. Depletion of Chp1 reduced expression of eEF1A but somewhat surprisingly had no overt growth phenotype like deleting TEF1. Also unexpectedly, CHP1 deletion did not cause major global changes in translation, although it did induce a stress response driven by Hsf1. This response was attributed to accumulation of misfolded eEF1A overwhelming the UPS system rather than a limited abundance of functional eEF1A. Accordingly, deleting TEF1 in cells lacking Chp1 reduced this stress, while an extra copy of TEF1 enhanced it (Fig 4c-e). Chp1 binds and stabilizes eEF1A with a mutation (F98C) analogous to a disease associated mutation in the human eEF1a counterpart.

The paper has broad interest, reports the discovery of an unknown ribosome-associated chaperone protein that is conserved in humans, and uncovers eEF1A as its primary substrate whose folding is dependent on its function. The study is thorough with a variety of genetic, biochemical and genomic approaches. The experiments are well-controlled and conclusions are supported by the data.

Comments:

The authors present much data and for most of it their descriptions and interpretations are very succinct and reasonably clear. One area that could use some clarifying would be on the relationship, or lack thereof, of the phenotypes of chp1 and tef1 cells:

- Deleting CHP1 does not affect cell doubling time but causes UPS stress and activates Hsf1.
- Deleting TEF1 reduces growth at 30 and 37 and reduces Hsf1 activity.
- increasing TEF1 does not affect growth but activates Hsf1.
- chp1 cells have 30% less eEF1A.

So, in chp1 cells the burden on UPS caused by misfolded eEF1A is not large enough to affect growth considerably, while the growth defect of tef1 cells seems due to a limited availability of eEF1A.

As cells lacking TEF1 have obvious growth defects it seems surprising that in *chp1* cells so much misfolded eEF1A is produced that it overwhelms the UPS system, yet there is no effect on global translation or a significant effect on doubling time, even at elevated temperature. If the growth defect of *tef1* cells is due to a reduction in eEF1A, then despite the 30% reduction of eEF1a in *chp1* cells, one would expect that this reduction of functional eEF1A in *chp1* cells to be far less than that in *tef1* cells. One might also conclude that the role of Chp1 in eEF1a biogenesis is not very important with regard to cell growth.

It would be helpful to assess whether the contrasting phenotypes are related to a difference in amount of eEF1A. Simply compare amounts of eEF1A in *chp1* and *tef1* cells. (It also might help to know how much EF1A must be reduced before it becomes limiting for growth.) If eEF1A is not much less abundant in *tef1* cells than in *chp1* cells, then deleting TEF1 must have detrimental effects unrelated to reduced abundance of eEF1A. Many proteins involved in translation and protein quality control were identified as interactors of Chp1. It could be edifying to identify genes among them that show true synthetic lethality with the *chp1* null mutant and the underlying cause of the interdependency.

The authors have the opportunity to put a name on ORF YLP225W and why they chose the non-descript "chaperone one" (Chp1) is a bit of a wonder.

Reviewer #3 (Remarks to the Author):

This is a very interesting work that identifies a new chaperone dedicated to the folding and quality control of eEF1A, an abundant and essential protein in eukaryotic cells. Using a combination of genetic screen, selective ribosome profiling, and protein interaction studies, the authors show that Chp1 (i) is selectively recruited, with the help of NAC, to the G-domain of eEF1A during its synthesis on the ribosome; (ii) interacts with the GTPase domain of eEF1A and recognizes eEF1A in non-native conformations; (iii) its absence impairs the de novo folding of eEF1A, leads to degradation of the newly synthesized eEF1A, and activates the Hsf1 stress response pathway. These results identify Chp1 as a dedicated chaperone that ensures the proper biogenesis of eEF1A, analogous to another recently discovered chaperone Zpr1, and demonstrates its importance in diseases rooted in eEF1A misfolding. Overall, this is a well-executed study that highlight the complexity of multi-domain protein folding and the importance of cotranslational chaperone interactions in ensuring this folding. It also provides another example of the role of NAC in recruiting protein biogenesis factors to nascent chains on the ribosome.

There are nevertheless a number of unresolved issues in this study that need to be addressed. These are detailed below:

1. p4, 2nd paragraph: “co-expression of Chp1 resulted in increased total and soluble levels of the eEF1A domain”. From the gel, it looks to me that eEF1A is well expressed and most of it is soluble even without Chp1 coexpression. It is not obvious that co-expression with Chp1 increased the yield or solubility. Does the author quantify this effect (with replicates)?

2. In Figure S3, the authors compared the interaction of Chp1 with misfolded vs native eEF1A and claimed that the latter is significantly weaker. It is not clear how interaction with ‘native eEF1A’ was measured, nor was the actual data presented. It appears that the assays and constructs used for the two measurements were different. Given the importance of this point in the manuscript, it will be prudent to compare the two interactions using the same assays and show the original data.

3. In general, I find it hard to reconcile the specificity of Chp1 interaction with ‘misfolded’ eEF1A and its stable interaction with the apparently stable, soluble, and likely well-folded eEF1A G-domain in *E. coli*. This also contradicts with the model in Figure 6, in which Chp1 only interacts with incompletely synthesized and unfolded G-domain. Do the authors have an explanation for this?

4. Part of the supporting evidence for Chp1-eEF1A interaction is the Alpha-fold multimer prediction, which comes with pLDDT values (confidence scores). The authors should show the confidence evaluations of the predictions and describe in more detail how the AlphaFold modeling was done. Conceptually, I assume AlphaFold predictors work on folded proteins / domains, which the author claims is not recognized by Chp1. I am not aware of predictors for protein interaction with unfolded protein or partial folding intermediates. It would be good to see how the authors did this modeling.

5. p5: Chp1 deletion reduced the steady state level of eEF1A by only 30% but its effect on eEF1A synthesis appears much larger (almost 3-fold). Can the authors explain why?

6. Fig. 3a: It is surprising that over-expression of TEF1 rescued cell growth in Δ chp1 cells given the likely misfolding of TEF1 in the absence of Chp1. Do the authors have an explanation?

7. Step 5 in Figure 6 is remarkably specific about the mechanism of Zpr1. What is the evidence and reasoning that support this mechanism?

8. NAC appears to aid in Chp1 recruitment to ribosomes but is not essential, and certainly plays a rather weak role in eEF1A biogenesis. The authors may want to be more circumspect in discussions about the role of NAC.

Point-to-point address to reviewer comments

Reviewer #1 (Remarks to the Author):

In this manuscript, Minoia et al. investigated a conserved and previously uncharacterized ribosome-associated factor (YPL225w), the absence of which triggers heat shock factor 1 (Hsf1) activation in yeast. The authors termed this factor chaperone 1 (CHP1) and found that it interacts with ribosomes that translate the protein eEF1A, a key translation elongation factor and one of the most abundant cellular proteins. Specifically, CHP1 binds to ribosomes when the N-terminal GTPase domain of eEF1A leaves the ribosomal tunnel suggesting that CHP1 helps to fold this domain during translation. The authors also provide evidence that knockout of CHP1 in yeast partially impairs eEF1A biogenesis and causes proteostasis stress. Additionally, the authors show an interaction between CHP1 and the N-terminus of eEF1A in a *E. coli* co-overexpression system. Based on these data, the authors conclude that CHP1 is a dedicated cotranslational chaperone for eEF1A.

Furthermore, the authors show that CHP1 interacts with the nascent polypeptide-associated complex (NAC), an abundant and ubiquitous heterodimeric complex located at the ribosome tunnel exit. NAC seems to stabilize ribosome binding of CHP1 through a direct interaction with the alpha-subunit of NAC. However, eEF1A biogenesis was not affected in the absence of NAC in yeast.

The discovery of a potential specific cotranslational chaperone for eEF1A that interacts with NAC is novel and would be of great interest to a wide readership. However, the presented data are too preliminary to provide a clear mechanistic model for the interaction of CHP1 with NAC and nascent eEF1A on ribosomes. For unknown reasons, obvious and simple experiments addressing the molecular interactions of CHP1 with eEF1A and NAC were not performed, and some conducted experiments appear to have technical flaws (see specific points below). The manuscript requires a major revision before I can support publication in a high-impact journal such as *Nat Commun*.

We thank the reviewer for recognizing that our study is of interest to a wide readership. During the revision we have addressed the raised concerns (see below).

Major points:

1. The authors need to provide clear evidence that the alpha-subunit of NAC interacts with CHP1. Based on the author's ColabFold model (Fig. S1e), alphaNAC contacts CHP1 via the C-terminal UBA domain. For unknown reasons, this was not mentioned in the text and not addressed experimentally. The authors should perform interactions studies *in vitro* and *in vivo* with NAC variants lacking the UBA domain. In addition, they should also mutate the UBA binding interface on CHP1.

We have performed a new *in vitro* experiment using purified Chp1 and NAC that demonstrate that Chp1 binds NAC and that this interaction is dependent on the presence of the UBA domain of α -

NAC/Egd2. This new data has been added as Fig. 1d. The Results (p. 4) have been updated to include the following description of the new data:

“Using purified components, we confirmed a direct interaction between Chp1 and NAC that was dependent on the presence of the UBA domain (Fig. 1d). Hence, Chp1 directly interacts with NAC via Egd2 dependent on the UBA domain.”

In a parallel approach, we deleted the sequences encoding the UBA domain of Egd2 in yeast cells, which resulted in complete loss of Egd2 expression. This behavior is consistent with what has been described by Panasencko et al. 2006, JBC and precludes a simple coIP experiment test using this construct.

In the revised manuscript we provide the following evidence for Chp1-NAC interaction: (1) Identification of Egd2 as a high-score Chp1 interactor by tandem affinity purification and MS analysis (Supplementary Data 1), (2) CoIP of Chp1-NAC from yeast cells (WT, *egd1Δ*, *egd2Δ*) (Fig. 1c), (3) Chp1-L12Bpa crosslinking to NAC *in vivo* (Supplementary Fig. 1c), (4) Reconstitution of the Chp1-NAC complex in *E.coli* (Supplementary Fig. 1e), (5) Coelution of Egd2 and Chp1 from ribosome pellets by 50 mM K⁺ (Fig. 1e), (6) Purified Chp1 and NAC form a complex dependent on the UBA domain (Fig. 1d).

2. Related to the previous point, the photo-crosslinking analysis in Fig. S1c is questionable: the authors used an antibody that detects both NAC subunits (alpha and beta). How can the authors be sure that the crosslink is to alphaNAC and not betaNAC? There seems to be an additional, less prominent NAC crosslink above the marked band, which, considering the size difference between alpha- and betaNAC, could be a crosslink to alphaNAC, while the stronger, marked crosslink is a betaNAC crosslink. The authors need to clarify this.

We thank the reviewer for pointing this out. The Results (p. 4) have been updated to include the following more careful description of the data:

“The α -subunit of NAC, Egd2, was ranked as a top hit and using serum detecting Egd1 and Egd2 we found that NAC directly crosslinked with Chp1-L12Bpa (Supplementary Fig. 1c).”

In light of the other experiments and structural models, we feel confident that the Chp1 interaction involves alphaNAC. These experiments include support from tandem-affinity purification and peptide mass fingerprinting identification using LC-MS on Chp1-L12Bpa in yeast that identified Egd2 with high score.

3. The structure of the central heterodimerization domain of yeast NAC in the author’s ColabFold model (Fig. S1e) is inconsistent with published structures of human NAC (PMID: 37347872, PMID: 35201867). The authors need to explain this. Is yeast NAC different or is just the model wrong? The authors should provide AlphaFold2 models of the NAC-CHP1 interaction of both, human and yeast, and compare the structures.

We have performed ColabFold modeling of both the human and yeast complexes. Both models support an interaction between Chp1 and the UBA domain of α -NAC, yet as pointed out by the reviewer, the models do not recapitulate all 12 strands of the β -barrel-like heterodimer (6:4 and

5:4 β -strand symmetry for the human and yeast proteins, respectively). The Results (p. 4) have been updated to include the following description of the models:

“We performed structural modeling of a complex between Chp1 and NAC and the human homologues using ColabFold. Despite that the models did not recapitulate all 12 strands of the β -barrel-like heterodimer determined by crystallography for the human NAC heterodimerization domain^{21,22} they both suggested that Chp1 binds NAC via direct interaction with the UBA domain of its α -subunit, Egd2 (Supplementary Fig. 1d and Supplementary Data 2)²³.”

The figure legend for Supplementary Fig. 1d has been updated to clearly point out the limitations of the models regarding the heterodimerization domain:

“Note that the heterodimerization domain structurally determined by crystallography is arranged in a head-to-head manner that generates a 12-stranded β -barrel-like heterodimer with six strands in each of the two major β -sheets, while the models display 6:4 and 5:4 β -strand symmetry for the human and yeast proteins, respectively.”

4. The authors demonstrate that the N-terminal domain of eEF1A is bound by CHP1. However, the authors did not experimentally investigate the substrate binding site of CHP1. According to the author’s ColabFold model, a N-terminal helix of CHP1 interacts with eEF1A. This model needs to be tested by mutational approaches (deletion and point mutants of CHP1), both *in vitro* and *in vivo*.

We have performed new site-specific crosslinking using the zero spacer Bpa incorporated at three independent amino acid position in the N-terminal α -helix of Chp1 to test the interaction with eEF1A domain I (new data in Supplementary Fig. 2c-d). This approach has been complemented with mutation of the N-terminal α -helix (new data in Supplementary Fig. 2e). The new data support that the N-terminal α -helix of Chp1 interacts with the domain I of eEF1A. The Results (p. 5) have been updated to include the following description of the data:

“Having established the Chp1-eEF1A domain I complex, we used *in vitro* site-specific UV crosslinking between purified Chp1 with single Bpa substitutions in the N-terminal α -helix (Q18, V21 or E25) and domain I of eEF1A to test the involvement of this helix in the interaction (Supplementary Fig. 2c). All three positions harboring the zero-spacer crosslinker Bpa formed specific crosslinks with eEF1A domain I, demonstrating direct interaction with the domain (Supplementary Fig. 2d). Upon mutational analysis we found using the heterologous coexpression system that alanine substitutions of Chp1 residues predicted to interact with eEF1A, or complete removal of residues 2 to 28 significantly decreased the interaction (Supplementary Fig. 2e). The most prominent impairment was observed for the deletion mutant, resulting in 58 % reduction of the binding.”

5. For unknown reasons, the authors used a plethora of diverse affinity tags on CHP1 (GFP, mCherry, HA, FLAG, SUMO, etc), which appear to be located sometimes at the N-terminus and sometimes at the C-terminus. Considering the small size of CHP1 (~17 kDa), use of large tags like GFP and mCherry (both ~27 kDa) is dangerous and could result in artefacts. The cotranslational interaction studies *in vivo* was performed with a C-terminally tagged variant, CHP1-GFP. The

authors should therefore stick to CHP1 variants with a C-terminal tag for their *in vitro* interaction studies. According to the authors model, a N-terminal helix of CHP1 interact with eEF1A. Therefore, a large tag like mCherry on the N-terminus should be avoided. I am in particular concerned with the analysis in Fig. S3f, where a mCherry tag was fused to the N-terminus of CHP1. Here, the authors wanted to prove that CHP1 only interacts with unfolded eEF1A, but not with natively folded eEF1A. The authors need to rule out that the mCherry tag blocks the substrate interaction.

We apologize for the ambiguous labeling regarding the mCherry tag on Chp1 in Supplementary Fig. 3f. It was in fact appended to the C-terminus of Chp1 and we have now label the figure with “Chp1-mCherry”.

Regarding Chp1 interaction with unfolded eEF1A we have performed additional experiments using Chp1-Bpa crosslinking and find support for our original notion that Chp1 interacts with unfolded eEF1A (Supplementary Fig. 2g). The Results (p. 7) have been updated to include the following description of the new data:

“In a parallel approach, *in vitro* site-specific UV crosslinking between Chp1 with Bpa substitution at position E25 and eEF1A demonstrated impaired interaction between the proteins in the presence of nucleotide or high glycerol concentration (Supplementary Fig. 2g).”

Regarding the use of tagged Chp1, we have carefully gone through all experiments and made sure that we use C-terminal tags of GFP/mCherry or small peptides. The only exception involves the SUMO solubility tag that we have used for *E. coli* expression. Importantly, we observe interaction between SUMO-Chp1 and eEF1A variants as well as NAC. The structural models display a 12 residues long unstructured tail preceding the N-terminal α -helix of Chp1 placing the N-terminus at a distance from the interaction interfaces reported on here.

6. The authors often use an *E. coli* protein co-overexpression system for protein-protein interaction studies. Why these artificial overexpression conditions in bacteria were used by the authors is puzzling. It would be better to express and purify the proteins individually and then perform interaction studies at physiological relevant protein concentrations *in vitro* in the test tube.

We appreciate the reviewer’s comment, and have in the revised version included *in vitro* interaction assays of Chp1-NAC and Chp1- eEF1A domain I as well as *in vitro* crosslinking between Chp1 and eEF1A.

The new experiment demonstrates an interaction *in vitro* between purified Chp1 and the domain I of eEF1A (Fig. 2e). The Results (p. 7) have been updated to include the following description of the new data:

“We directly tested whether Chp1 and the isolated domain I of eEF1A associate. Indeed, we found that purified Chp1 and eEF1A domain I formed a complex *in vitro* (Fig. 2e).”

The *in vitro* experiment demonstrating Chp1-NAC interaction is described under the reviewer’s point 1 above.

The crosslinking experiment with purified Chp1 and eEF1A domain I is described under the reviewer's point 4 above.

The crosslinking experiment with purified Chp1 and eEF1A is described under the reviewer's point 5 above.

Regarding the use of *E. coli* coexpression, our experience is that this approach repeatedly has been a successful path for the identification of eukaryotic protein complexes as well as to provide a source for their biochemical isolation.

Minor points

1. The SEC analysis in Fig. S1d is questionable, because alphaNAC was expressed alone without betaNAC. This results in formation of artificial alphaNAC homodimers. These homodimers would have two CHP1 binding sites (e.g., two UBA domains). The authors should purify the heterodimeric NAC complex (with and without the UBA domain) and repeat this analysis.

We agree that the experiment is preliminary. We prefer to report on the analysis of Chp1 interaction with the proper NAC heterodimer. In light of the new data showing an interaction with NAC depending on the UBA domain using purified proteins (see major point 1 above) we have decided to remove this data from the manuscript.

2. The strongest evidence that eEF1A biogenesis is indeed impaired in CHP1 knockout cells is the analysis in Fig. 3f using the GAL-induced expression system (the differences in the eEF1A steady-state levels are very minor, and near the detection limit of immunoblotting ~30%). However, for the GAL system important controls are missing. The authors should induce expression of a control protein unrelated to CHP1 and/or provide mRNA levels of eEF1A.

We have determined the mRNA levels for eEF1A during the GAL-induced expression. The data have been incorporated into the updated figure Fig. 3e and show comparable induction. The Results (p. 8) have been updated:

“In contrast, when eEF1A was expressed from the induced *GAL1* promoter, the rate of eEF1A production was drastically reduced in *chp1Δ* cells, while mRNA levels were comparable in WT and *chp1Δ* cells, indicating that Chp1 plays a major role in specifically *de novo* protein synthesis of eEF1A (Fig. 3e).”

3. The authors have done themselves no favors by analyzing a cotranslational NAC-related mechanism in yeast, the only known organism, in which NAC is not essential. There are clear functional differences between NAC in yeast and higher eukaryotes. If possible, the authors should perform PBDC1 and NAC knockdown experiments in human cells and analyze potential defects in eEF1A biogenesis (which might be much stronger than in yeast!).

Although we understand that the yeast model has its limitations regarding the study of NAC function, what we offer regarding NAC is awareness of the interaction with Chp1, how NAC is required to maintain Chp1 bound to the nascent eEF1A domain I during its synthesis and the involvement of the UBA domain in the interaction with Chp1. We feel that these are important

contributions. Expanding the scope of the study to also involve work in human cells is beyond what is doable for us during this revision.

Reviewer #2 (Remarks to the Author):

To identify ribosome-associated chaperones, Minoia et al. compared ribosome-associated proteins in cells with different gene deletions that activate Hsf1. They discovered previously unknown ORF YPL225W among the strongest activators of Hsf1. They name the gene CHP1 for Chaperone 1. Using Bpa crosslinking they find that Chp1 binds NAC subunit Egd2, and that NAC helps recruit Chp1 to ribosomes. Using selective ribosome profiling they identify TEF1/2 (encoding eEF1A) as the most enriched transcripts, which led them to suggest eEF1A is the sole nascent chain substrate of Chp1. Findings from various assays support their conclusions that NAC facilitates binding of Chp1 to the N-terminal GTPase domain of eEF1A, which stabilizes folding of eEF1A as it emerges from the ribosome. Depletion of Chp1 reduced expression of eEF1A but somewhat surprisingly had no overt growth phenotype like deleting TEF1. Also unexpectedly, CHP1 deletion did not cause major global changes in translation, although it did induce a stress response driven by Hsf1. This response was attributed to accumulation of misfolded eEF1A overwhelming the UPS system rather than a limited abundance of functional eEF1A. Accordingly, deleting TEF1 in cells lacking Chp1 reduced this stress, while an extra copy of TEF1 enhanced it (Fig 4c-e). Chp1 binds and stabilizes eEF1A with a mutation (F98C) analogous to a disease associated mutation in the human eEF1a counterpart.

The paper has broad interest, reports the discovery of an unknown ribosome-associated chaperone protein that is conserved in humans, and uncovers eEF1A as its primary substrate whose folding is dependent on its function. The study is thorough with a variety of genetic, biochemical and genomic approaches. The experiments are well-controlled and conclusions are supported by the data.

We thank the reviewer for the positive review.

Comments:

The authors present much data and for most of it their descriptions and interpretations are very succinct and reasonably clear. One area that could use some clarifying would be on the relationship, or lack thereof, of the phenotypes of *chp1* and *tef1* cells:

- Deleting CHP1 does not affect cell doubling time but causes UPS stress and activates Hsf1.
- Deleting TEF1 reduces growth at 30 and 37 and reduces Hsf1 activity.
- increasing TEF1 does not affect growth but activates Hsf1.
- *chp1* cells have 30% less eEF1A.

So, in *chp1* cells the burden on UPS caused by misfolded eEF1A is not large enough to affect growth considerably, while the growth defect of *tef1* cells seems due to a limited availability of eEF1A.

As cells lacking TEF1 have obvious growth defects it seems surprising that in *chp1* cells so much misfolded eEF1A is produced that it overwhelms the UPS system, yet there is no effect on global translation or a significant effect on doubling time, even at elevated temperature. If the growth defect of *tef1* cells is due to a reduction in eEF1A, then despite the 30% reduction of eEF1a in *chp1* cells, one would expect that this reduction of functional eEF1A in *chp1* cells to be far less than that in *tef1* cells. One might also conclude that the role of Chp1 in eEF1a biogenesis is not very important with regard to cell growth.

It would be helpful to assess whether the contrasting phenotypes are related to a difference in amount of eEF1A. Simply compare amounts of eEF1A in *chp1* and *tef1* cells. (It also might help to know how much EF1A must be reduced before it becomes limiting for growth.) If eEF1A is not much less abundant in *tef1* cells than in *chp1* cells, then deleting TEF1 must have detrimental effects unrelated to reduced abundance of eEF1A. Many proteins involved in translation and protein quality control were identified as interactors of Chp1. It could be edifying to identify genes among them that show true synthetic lethality with the *chp1* null mutant and the underlying cause of the interdependency.

Spurred by the feedback from the reviewer, we have carefully quantified the levels of eEF1A in the various *TEF1* and *CHP1* mutants using highly reproducible near-IR-based western analysis. The new data have been added as Fig. 3c and support the interpretation that *tef1* Δ cells grow slower than *chp1* Δ cells due to lower levels of eEF1A. The Results (p. 8) have been updated to include the following description of the new data:

“We quantified eEF1A levels in the mutants using western blot analysis (Fig. 3c and Supplementary Fig. 5c). In line with the growth phenotypes, we observed 36% reduction of eEF1A levels in *chp1* Δ cells, while *tef1* Δ cells exhibited 74% reduction. Hence, the lower levels of eEF1A in *chp1* Δ cells result in only mild growth phenotypes, while further reduction of the levels in *tef1* Δ cells apparently makes translation elongation rates limiting for growth. The *chp1* Δ *tef1* Δ cells exhibited 78% decrease of eEF1A levels and further reduced growth compared to the *tef1* Δ strain. Interestingly, while +*TEF1* cells expressed 184% eEF1A, *chp1* Δ +*TEF1* cells failed to significantly increase the *eEF1A* levels compared to *chp1* Δ cells, suggesting that eEF1A expression requires Chp1.”

In regards to putative Chp1 interactors involved in translation and protein quality control obtained by MS analysis and their genetic interactions with *chp1* Δ (the CellMap), we have compared the physical interactors with the negative genetic interactors (Supplementary Fig. 4c). The data lend support to our findings. The Results (p. 7) have been updated to include the following description of the new data:

“This was further supported by finding that 7 of the *CHP1* negative interactors involved in translation and protein quality control (*YDJI*, J-domain protein for Hsp70; *TEF1*, eEF1A; *TEF4*, eEF1B γ ; *RSP5*, ubiquitin E3 ligase; *HYP2*, eIF5A; *RPS0A*, ribosomal 40S subunit protein S0A; *RPT1*, ATPase 19S proteasome) were also identified as physical interactors in our MS analysis (Supplementary Fig. 4c).”

The authors have the opportunity to put a name on ORF YLP225W and why they chose the non-descript “chaperone one” (Chp1) is a bit of a wonder.

Thanks for the suggestion, we have changed to a more precise description: Chaperone 1 for eEF1A (*CHP1*). We think this makes a lot of sense in light of that Chp1 acts very early in the folding of eEF1A. The Results now read:

“Interestingly the uncharacterized open reading frame (ORF) YPL225w, here named *CHP1* (Chaperone 1 for eEF1A), was associated with one of the strongest activations of Hsf1 among all the RAP.”

Reviewer #3 (Remarks to the Author):

This is a very interesting work that identifies a new chaperone dedicated to the folding and quality control of eEF1A, an abundant and essential protein in eukaryotic cells. Using a combination of genetic screen, selective ribosome profiling, and protein interaction studies, the authors show that Chp1 (i) is selectively recruited, with the help of NAC, to the G-domain of eEF1A during its synthesis on the ribosome; (ii) interacts with the GTPase domain of eEF1A and recognizes eEF1A in non-native conformations; (iii) its absence impairs the de novo folding of eEF1A, leads to degradation of the newly synthesized eEF1A, and activates the Hsf1 stress response pathway. These results identify Chp1 as a dedicated chaperone that ensures the proper biogenesis of eEF1A, analogous to another recently discovered chaperone Zpr1, and demonstrates its importance in diseases rooted in eEF1A misfolding. Overall, this is a well-executed study that highlight the complexity of multi-domain protein folding and the importance of cotranslational chaperone interactions in ensuring this folding. It also provides another example of the role of NAC in recruiting protein biogenesis factors to nascent chains on the ribosome.

We thank the reviewer for the positive review.

There are nevertheless a number of unresolved issues in this study that need to be addressed. These are detailed below:

1. p4, 2nd paragraph: “co-expression of Chp1 resulted in increased total and soluble levels of the eEF1A domain”. From the gel, it looks to me that eEF1A is well expressed and most of it is soluble even without Chp1 coexpression. It is not obvious that co-expression with Chp1 increased the yield or solubility. Does the author quantify this effect (with replicates)?

We agree with the reviewer that eEF1A domain I is expressed in *E. coli* as a soluble protein even in the absence of Chp1. What we observe are increased expression levels of this soluble protein when Chp1 is co-expressed. A representative gel is present in Fig. 2g and the quantifications of eEF1A domain I levels in the total and soluble fractions of triplicate experiments are shown in Figs. 2h and 2i, respectively.

In contrast, full length eEF1A quantitatively aggregates when expressed in *E. coli*. In this scenario Chp1 co-expression allows a fraction of eEF1A to become soluble but with no change in total expression levels (see Supplementary Fig. 3c-e)

We have clarified our description in the Results (p. 5):

“Interestingly, coexpression of Chp1 with the GTPase domain in *E. coli* cells increased the expression levels of the eEF1A domain (Fig. 2g-i).”

2. In Figure S3, the authors compared the interaction of Chp1 with misfolded vs native eEF1A and claimed that the latter is significantly weaker. It is not clear how interaction with ‘native eEF1A’ was measured, nor was the actual data presented. It appears that the assays and constructs used for the two measurements were different. Given the importance of this point in the manuscript, it will be prudent to compare the two interactions using the same assays and show the original data.

We apologize for the unclarity regarding the experiment and have now updated Supplementary Fig. 3f so that it is clear that we used copurification of 6xHis-eEF1A with Chp1-mCherry (anti-RFP nanobodies for coIP) to measure the interaction. A representative gel is now also included to complement the quantifications. We have also updated the description of the Results (p. 6-7) as follows:

“While Chp1 interacts with misfolded eEF1A, we predicted it should not bind the natively folded protein. To investigate this interaction, we performed an *in vitro* co-IP of Chp1-mCherry using eEF1A purified from its native yeast. Indeed, only minimal interaction between Chp1 and eEF1A could be observed with the highest interaction being detected in the presence of EDTA (10% of purified Chp1 was bound to eEF1A), a treatment that impedes nucleotide binding and thus destabilizes the GTPase domain¹⁵ (Supplementary Fig. 3f). Preincubation of eEF1A with nucleotide or high glycerol concentrations, both known to stabilize the protein^{15, 29, 30}, released Chp1.”

Based on the literature, incubation of eEF1A with high glycerol concentrations or nucleotide leads to the protein populating native conformations, hence we used these conditions to maximize the levels of natively folded eEF1A. In contrast, incubation of eEF1A with EDTA strips the nucleotide from eEF1A resulting in non-native conformations.

Encouraged by this comment and a comment from reviewer 1, we used these conditions in combination with photo crosslinking of eEF1A with Chp1E25Bpa. The new data supports the coIP results and have been incorporated as a new supplementary Fig. 3g. The Results (p. 7) have been updated to include the following description of the new data:

“In a parallel approach, *in vitro* site-specific UV crosslinking between Chp1 with Bpa substitution at position E25 and eEF1A demonstrated impaired interaction between the proteins in the presence of nucleotide or high glycerol concentration (Supplementary Fig. 3g).”

3. In general, I find it hard to reconcile the specificity of Chp1 interaction with ‘misfolded’ eEF1A and its stable interaction with the apparently stable, soluble, and likely well-folded eEF1A G-domain in *E. coli*. This also contradicts with the model in Figure 6, in which Chp1 only interacts

with incompletely synthesized and unfolded G-domain. Do the authors have an explanation for this?

Our understanding is that the isolated GTPase domain of eEF1A is soluble yet, exhibits conformational flexibility allowing it to adopt the specific nucleotide-free conformation that Chp1 binds, hence a specific non-native (“misfolded”) conformation. Chp1 binding results in increased expression of the GTPase domain likely by stabilizing it. We have updated the Results (p. 6) to make our reasoning clearer:

“Thus, the cotranslational interactions between Chp1 and the eEF1A GTPase domain that we detected by the SeRP are recapitulated in the ColabFold model as well as by reconstitution. According to the model, this interaction is not compatible with a nucleotide-bound conformation of the eEF1A GTPase domain. Moreover, Chp1 forms a stable complex with the GTPase domain and increases its expression level in a heterologous host. Together this data suggest that the stability of the domain increases when bound by Chp1.”

Thus, we find that the model in Fig. 6 is in agreement with the data.

4. Part of the supporting evidence for Chp1-eEF1A interaction is the Alpha-fold multimer prediction, which comes with pLDDT values (confidence scores). The authors should show the confidence evaluations of the predictions and describe in more detail how the AlphaFold modeling was done. Conceptually, I assume AlphaFold predictors work on folded proteins / domains, which the author claims is not recognized by Chp1. I am not aware of predictors for protein interaction with unfolded protein or partial folding intermediates. It would be good to see how the authors did this modeling.

We have updated the Methods section with detailed descriptions on how the ColabFold runs were executed. Model selection was done by average pLDDT values for each model.

ColabFold predicted a specific conformation of the GTPase domain of eEF1A that Chp1 binds. This conformation is not compatible with nucleotide binding and is not representative of the active conformation. In the updated manuscript we have provided experimental validation of this model including testing of amino acid residues of Chp1 involved in the interaction extending our original data that Chp1 recognizes amino acids 1 to 70 of eEF1A (see new Fig. 2e, Fig. 2f, Fig. 2j and new Supplementary Figs. 2d-e). Thus, our data show that Chp1 binds a specific conformation of the fragment $\beta 1$ - $\alpha 1$ - $\alpha 2$ - $\alpha 3$ (aa 1-70) of the GTPase domain. The Results (p 4) have been updated to include the following description of the new data:

“Having established the Chp1-eEF1A domain I complex, we used *in vitro* site-specific UV crosslinking between purified Chp1 with single Bpa substitutions in the N-terminal α -helix (Q18, V21 or E25) and domain I of eEF1A to test the involvement of this helix in the interaction (Supplementary Fig. 2c). All three positions harboring the zero-spacer crosslinker Bpa formed specific crosslinks with eEF1A domain I, demonstrating direct interaction with the domain (Supplementary Fig. 2d). Upon mutational analysis we found using the heterologous coexpression system that alanine substitutions of Chp1 residues predicted to interact with eEF1A, or complete removal of residues 2 to 28 significantly decreased the interaction (Supplementary Fig. 2e). The

most prominent impairment was observed for the deletion mutant, resulting in 58 % reduction of the binding.”

The Discussion (p. 11) has been updated to reflect that Chp1 binds a specific conformation of eEF1A domain I during its biogenesis:

“We find that Chp1 interacts with ribosome-eEF1A nascent chain complexes during the synthesis of its GTPase domain. SeRP and biochemical analysis indicate that Chp1 selectively binds the nascent GTPase domain of eEF1A cotranslationally as soon as the first 70 amino acids of the protein have emerged at the polypeptide tunnel exit. Chp1 uses its N-terminal α -helix to bind the β 1- α 1- α 2- α 3 of eEF1A in a specific conformation. Coexpression of the isolated GTPase domain and Chp1 in a heterologous *E. coli* system suggests that Chp1 interaction with the domain increases its stability and potentially protects it from degradation. Exposure of the complete GTPase domain at the ribosomal tunnel exit triggers the dissociation of Chp1, likely due to cotranslational folding of the domain into its stable nucleotide-bound structure.”

5. p5: Chp1 deletion reduced the steady state level of eEF1A by only 30% but its effect on eEF1A synthesis appears much larger (almost 3-fold). Can the authors explain why?

We have performed a new experiment that likely explains this discrepancy by linking it to the expression rates of eEF1A (Fig. 3c). Upon carefully assessing the steady state levels of eEF1A using western blot analysis, we find a larger relative reduction of eEF1A levels in *chp1* Δ +*TEF1* with respect to +*TEF1* than what is observed in *chp1* Δ with respect to WT. Thus, when using the *GALI* promoter to transiently express eEF1A we likely have an overexpression condition that accentuates the need for Chp1 compared to the expression from the endogenous promoters. The Results (p. 8) have been updated as follows to describe new Fig. 3c:

“We quantified eEF1A levels in the mutants using western blot analysis (Fig. 3c and Supplementary Fig. 5c). In line with the growth phenotypes, we observed 36% reduction of eEF1A levels in *chp1* Δ cells, while *tef1* Δ cells exhibited 74% reduction. Hence, the lower levels of eEF1A in *chp1* Δ cells result in only mild growth phenotypes, while further reduction of the levels in *tef1* Δ cells apparently makes translation elongation rates limiting for growth. The *chp1* Δ *tef1* Δ cells exhibited 78% decrease of eEF1A levels and further reduced growth compared to the *tef1* Δ strain. Interestingly, while +*TEF1* cells expressed 184% eEF1A, *chp1* Δ +*TEF1* cells failed to significantly increase the *eEF1A* levels compared to *chp1* Δ cells, suggesting that eEF1A expression requires Chp1. Thus, overexpression of eEF1A increased the dependency on Chp1 for its successful production.”

The Discussion (p. 12) has been updated to clarify our line of reasoning:

“Moreover, the *chp1* Δ phenotype regarding eEF1A expression becomes accentuated upon eEF1A overexpression, suggesting that downstream components, perhaps Zpr1-Aim29, fail to handle the increased synthesis rates.”

6. Fig. 3a: It is surprising that over-expression of TEF1 rescued cell growth in Δ chp1 cells given the likely misfolding of TEF1 in the absence of Chp1. Do the authors have an explanation?

To clarify, the doubling times of logarithmic growing cells are comparable between *chp1* Δ and *chp1* Δ +*TEF1* at both 30 and 37 °C (Fig. 3b and Supplementary Fig. 5b). At the stressful temperature of 39 °C we observed a severe growth retardation of specifically the *chp1* Δ +*TEF1* (Supplementary Fig. 7a). The only rescue we observe is suppression of the diauxic shift at lower cell densities of *chp1* Δ cells (Fig. 3a), a phenotype that likely involves complex glucose signaling. The Results (p. 7) have been updated to clarify that there is no change in doubling time:

“Also, *chp1* Δ cells displayed a mild growth impairment most clearly visible at 37 °C in the form of diauxic shift at lower cell densities but with no impact on doubling times. Introducing an extra copy of the *TEF1* gene (+*TEF1*) did not visibly impact the growth of the WT cells but suppressed the diauxic shift phenotype of *chp1* Δ cells. Thus, decreasing the gene dosage of eEF1A makes the maximal growth potential of cells sensitive to the loss of Chp1.”

7. Step 5 in Figure 6 is remarkably specific about the mechanism of Zpr1. What is the evidence and reasoning that support this mechanism?

We have updated the graphical model to reflect the literature on Zpr1-Aim29. References are present in the Introduction, the first paragraph of the Discussion and the legend of Fig. 6.

8. NAC appears to aid in Chp1 recruitment to ribosomes but is not essential, and certainly plays a rather weak role in eEF1A biogenesis. The authors may want to be more circumspect in discussions about the role of NAC.

We have clarified the Discussion (p. 12) to reflect our findings regarding NAC function in eEF1A biogenesis:

“Nevertheless, inactivation of NAC does not result in decreased eEF1A levels raising the possibility that Chp1-NAC interaction does not directly facilitate eEF1A biogenesis.”

REVIEWERS' COMMENTS

Reviewer #1 (Remarks to the Author):

The authors have addressed all points in a satisfying way. How exactly Chp1 interacts with nascent eEF1A and what this implies is not yet clear, but the study identifies a new cotranslational eEF1a-specific factor which is novel and exciting and should therefore be accepted for publication.

Enclosed are a few minor points that should be improved prior publication.

1. Page 5: "In addition, size-exclusion chromatography analysis of the copurified fraction from the E. coli recombinant system confirmed the presence of a stable Chp1-eEF1A domain I complex with an apparent stoichiometry of 1:1 (Supplementary Fig. 2b)."

How can a 1:1 stoichiometry be derived from this figure?

2. Page 4, line 113

Reference 24 refers to no sentence.

3. CHP1 or Chp1? Stay consistent in the manuscript.

Reviewer #2 (Remarks to the Author):

My major criticism of the paper by Minoia et al. was the need to explain why growth phenotypes cells lacking Chp1, which show a significant reduction in eEF1A and overburdened UPS, were not nearly as bad as those lacking Tef1. The authors quantified the amounts of eEF1A in the mutant cells and find those lacking Tef1 have considerably less eEF1A than those lacking Chp1 (Fig 3c). The new data suggest a simple explanation and satisfy my concerns.

Reviewer #3 (Remarks to the Author):

In this revision, the authors have addressed most of my questions and comments from the previous round. The model that Chp1 cotranslationally stabilizes a near-native but nevertheless disordered state of eEF1 G-domain provides a consistent explanation for the data presented in this work. I support publication.

Point-to-point address

REVIEWERS' COMMENTS

Reviewer #1 (Remarks to the Author):

The authors have addressed all points in a satisfying way. How exactly Chp1 interacts with nascent eEF1A and what this implies is not yet clear, but the study identifies a new cotranslational eEF1a-specific factor which is novel and exciting and should therefore be accepted for publication.

We thank the reviewer for the careful assessment.

Enclosed are a few minor points that should be improved prior publication.

1. Page 5: "In addition, size-exclusion chromatography analysis of the copurified fraction from the E. coli recombinant system confirmed the presence of a stable Chp1-eEF1A domain I complex with an apparent stoichiometry of 1:1 (Supplementary Fig. 2b)."

How can a 1:1 stoichiometry be derived from this figure?

The reviewer is correct and we have removed the statement of stoichiometry relating to this figure panel. Nevertheless, several other lines of evidence in the study indicate a 1:1 stoichiometry and we have represented this in the graphical model (Fig. 6).

2. Page 4, line 113

Reference 24 refers to no sentence.

The reference has been removed.

3. CHP1 or Chp1? Stay consistent in the manuscript.

In the manuscript, we use standard nomenclature for indicating genes and proteins derived from yeast, i.e. *CHP1* [it.] indicates the gene and Chp1 indicates the protein. We have carefully gone through the text and figures and each usage is correct.

Reviewer #2 (Remarks to the Author):

My major criticism of the paper by Minoia et al. was the need to explain why growth phenotypes cells lacking Chp1, which show a significant reduction in eEF1A and overburdened UPS, were not nearly as bad as those lacking Tef1. The authors quantified the amounts of eEF1A in the mutant cells and find those lacking Tef1 have considerably less eEF1A than those lacking Chp1 (Fig 3c). The new data suggest a simple explanation and satisfy my concerns.

We thank the reviewer for supporting publication.

Reviewer #3 (Remarks to the Author):

In this revision, the authors have addressed most of my questions and comments from the previous round. The model that Chp1 cotranslationally stabilizes a near-native but nevertheless disordered state of eEF1 G-domain provides a consistent explanation for the data presented in this work. I support publication.

We thank the reviewer for the support.